# New Directions to Approach Oxidative Stress Related to Physical Activity and Nutraceuticals in Normal Aging and Neurodegenerative Aging

**DOI:** 10.3390/antiox12051008

**Published:** 2023-04-26

**Authors:** Manuela Violeta Bacanoiu, Mircea Danoiu, Ligia Rusu, Mihnea Ion Marin

**Affiliations:** 1Sport Medicine and Physical Therapy Department, Faculty of Physical Education and Sport, University of Craiova, 200585 Craiova, Romania; manuela.bacanoiu@edu.ucv.ro (M.V.B.); mircea.danoiu@edu.ucv.ro (M.D.); 2Faculty of Mechanics, University of Craiova, 200585 Craiova, Romania; mihnea.marin@edu.ucv.ro

**Keywords:** oxidative stress, older adults, normal aging, physical activity, neurodegenerative pathology aging, redox potential

## Abstract

Oxidative stress (OS) plays, perhaps, the most important role in the advanced aging process, cognitive impairment and pathogenesis of neurodegenerative disorders. The process generates tissue damage via specific mechanisms on proteins, lipids and nucleic acids of the cells. An imbalance between the excessive production of oxygen- and nitrogen-reactive species and antioxidants leads to a progressive decline in physiological, biological and cognitive functions. Accordingly, we need to design and develop favourable strategies for stopping the early aging process as well as the development of neurodegenerative diseases. Exercise training and natural or artificial nutraceutical intake are considered therapeutic interventions that reduce the inflammatory process, increase antioxidant capacities and promote healthy aging by decreasing the amount of reactive oxygen species (ROS). The aim of our review is to present research results in the field of oxidative stress related to physical activity and nutraceutical administration for the improvement of the aging process, but also related to reducing the neurodegeneration process based on analysing the beneficial effects of several antioxidants, such as physical activity, artificial and natural nutraceuticals, as well as the tools by which they are evaluated. In this paper, we assess the recent findings in the field of oxidative stress by analysing intervention antioxidants, anti-inflammatory markers and physical activity in healthy older adults and the elderly population with dementia and Parkinson’s disease. By searching for studies from the last few years, we observed new trends for approaching the reduction in redox potential using different tools that evaluate regular physical activity, as well as antioxidant and anti-inflammatory markers preventing premature aging and the progress of disabilities in neurodegenerative diseases. The results of our review show that regular physical activity, supplemented with vitamins and oligomolecules, results in a decrease in IL-6 and an increase in IL-10, and has an influence on the oxidative metabolism capacity. In conclusion, physical activity provides an antioxidant-protective effect by decreasing free radicals and proinflammatory markers.

## 1. Introduction

Concern for a healthy lifestyle in older adults and delaying the evolution of deficiencies in neurodegenerative diseases represent two of the most important problems in healthy people. The decline that accompanies normal aging and pathological aging involves socioeconomic and psychosocial determinants that must be limited. Elderly people’s quality of life deteriorates owing to the lowering of motor functions and mental health, problems that require solutions. Improving these functions not only requires financial interventions both in specialised medical units and at home, but also the active involvement of health policies with governmental participation. From this point of view, the independence of elderly people in apparent good health or with neurodegenerative disabilities must be stimulated, personal identity needs to be strengthened and confidence in participating in independent daily activities must be increased. In the last decade, premature aging was one of the most important concerns because this phenomenon is accompanied by a decline in physical functions and increased cognitive impairment. At the same time, quality of life and wellbeing are affected, and the recovery and treatment procedures require rehabilitation, institutionalisation and hospitalisation, with large financial implications. Thus, the advancement in age with the related decline has prompted massive research due to a need for new strategies that allow as much delay as possible in the alteration of elderly adults’ state of health. Age-related morphological and functional changes induce changes in mobility, balance and joint flexibility, and increase the risk of falls, as well as functional cognitive decline. Therefore, aging is a biological and irreversible process that is often accompanied by multiple comorbidities, especially neurodegenerative diseases, examples being dementia and Alzheimer’s disease. Accordingly, the impairment of physical and intellectual capacities leads to an alteration in daily activities in terms of wellbeing and quality of life and decreases independent living capability. The hallmarks of aging include reduced physical capacities, altered cognitive status and, at the infrastructure level, disrupted balance in homeostasis, genomic stability and mitochondrial functionality. Since 1956, aging has been associated with oxidative stress, which is defined as an alteration balance between oxidant and antioxidant biomolecules [1]. There are several factors that contribute to the aging process, such as a sedentary lifestyle, poor diet, loss of intellectual activity and an induced altered redox status. Continuous functional fitness activities and low-, moderate- or high-intensity physical exercises improve the gait pattern, strength and resistance of muscles, balance, flexibility of joints and the mobility in movements. In this regard, physical interventions represent a preventive strategy for avoiding chronic oxidative stress [2]. On the other hand, changes in variables such as lifestyle, behavioural habits, intake of natural and artificial nutraceuticals and anti-inflammatory compounds succeed in lowering oxidant agents and bio-inflammatory markers in healthy older adults or seniors with neurodegenerative disorders.

## 2. Methods of the Literature Review

In order to create a narrative review, we browsed the PubMed database with the keywords: “oxidative stress”, “older adults”, “normal aging”, “physical activity”, “aging and neurodegenerative disorders” and “nutraceuticals”. Our research includes original articles, review articles and case reports from the last five to six years. All original articles and clinical trials, regarding physical training, functional fitness or supplementation with natural and artificial antioxidants, demonstrated a decrease in oxygen- or nitrogen-reactive species and inflammatory markers in elderly people. By applying these strategies, the importance of physical training and positive nutraceutical intervention was proven to delay premature aging as well as the progress of neurodegenerative diseases, thereby influencing quality of life or wellbeing. Finally, only the most relevant studies, written in English, were selected from the reference list. Our review proposes to present information about the following important topics: oxidative stress and aging pathophysiology, the impact of physical activity and nutraceutical compounds on oxidative stress for healthy aging, the impact of physical activity and oxidative stress for older adults and the impact of recovery methods using physical activity and nutraceutical compounds on oxidative stress for neurodegenerative diseases.

## 3. Oxidative Stress and Aging Process Pathophysiology

Oxidative stress is the result of an imbalance between excessive production of reactive oxygen species and reactive nitrogen species (ROS/RNS) and an antioxidant defence that cannot neutralise them. Free radicals are markers of oxidative stress and are atoms or molecules with one or more than one unpaired electron in an external shell. They are formed from the interaction between oxygen and certain molecules [3].

Reactive oxygen species (ROS) and reactive nitrogen species (RNS) are two species of reactive oxygen, both produced by aerobic cells and associated with the aging process, as well as age-related diseases [4]. 

RONS (reactive oxygen and nitrogen species) have endogenous sources, including nicotinamide adenine dinucleotide phosphate (NADPH) oxidase, myeloperoxidase (MPO), lipoxygenase and angiotensin II, as well as exogenous sources such as air and water pollution, tobacco, alcohol, heavy or transition metals, drugs, industrial solvents, cooking and radiation, all of which result in metabolization into free radicals inside the body [5].

Methods for the assessment of oxidative stress are focused on the direct measurement of reactive oxygen species—responsible for the deleterious effects of oxidative stress—using fluorogenic probes [6]. The indirect measurement of ROS is also a method based on the analysis of the oxidative damage these radicals cause to lipids (lipid peroxidation, malondialdehyde and oxidised levels of low-density lipoproteins), proteins (protein carbonyl and the detection of advanced oxidation protein products) and cell nucleic acids (8-hydroxy-2′-deoxyguanosine, and thymidine glycol, which is a specific marker for oxidative DNA damage) [7].

Regarding the treatment of oxidative stress, the administration of antioxidant molecules, including nutritional supplements such as vitamin A (which modifies the effect of apolipoproteins on the risk of myocardial infarction), is the recommended treatment [8]. Vitamin C can control endothelial cell proliferation and apoptosis and smooth muscle-mediated vasodilation. Novel omega-3-based antioxidants and novel therapeutic strategies using miRNA and nanomedicine are also being developed [9].

Oxidative stress alters macromolecules in the body, such as lipids, proteins and nucleic acids. Finally, ROS and RNS can lead to chronic inflammation and to the activation of pathological metabolic pathways that cause illness. Redox stress/oxidative stress is a complex process. Its impact on the body depends on the type of oxidant, the place and intensity of its production, the composition, the activities of various enzymatic or nonenzymatic antioxidants and the capacity of self-repair systems. The oxidative stress theory is currently the most accepted explanation for aging, stipulating that excess ROS leads to genetic, molecular, cell, tissue and systemic changes—the level of reactive oxygen species (ROS) increases and is associated with an increased risk of tissue damage. This increase in oxidative activity generates impairments in nucleic acids (DNA/RNA), lipids and proteins, and affects metabolic and biochemical pathways, modifying cell homeostasis, signal transduction and gene expression. Therefore, in the process of cell respiration in the mitochondria, organic peroxides, reactive aldehydes such as malondialdehyde (MDA) and nitric oxide—which generates pathological modifications in proteins, lipids and DNA/RNA and increases the chance of genetic mutations—are produced [2]. Endogenous antioxidants include superoxide dismutase (SOD), catalase (CAT), glutathione reductase (GSR) and glutathione peroxidase (GPX) as a defence system. They cannot counteract the oxidizing action of free radicals responsible for the appearance of heat-shock proteins (HSPs), inflammatory markers that ultimately lead to cell apoptosis [10]. Oxidised proteins produce protein carbonyl (CP) via lipid peroxidation related to isoprostanes (8-iso-PGF2α) or produce glycated compounds that lead to neuron degeneration, damage synaptic networks, increase neuroinflammation and result in subsequent memory loss with the onset and progression of neurodegenerative diseases [11]. Oxidative stress can activate a variety of proinflammatory factors, such as the tumour necrosis factor alpha (TNF-α), interleukins IL-6 and IL-8, thiobarbituric acid-reactive substances (TBARS) and homocysteine (Hcy), all of which are associated with a brain-level neuroinflammation response [12,13]. Oxidative stress in the brain is mostly due to the production of the factor insoluble amyloid plaques (Aβ42), oligomers that stimulate hyperphosphorylation of tau proteins (p-tau). Neuroinflammation is a key player in Alzheimer’s disease (AD), with the active participation of proinflammatory cytokines such as IL-1β and TNF-α, which promote neuronal and synaptic disorders [14]. For healthy aging and delaying the evolution of the pathological neurodegenerative process in elderly people, mechanisms must be promoted to adjust the imbalance between the excessive production of ROS and the decrease in the defensive enzyme system and, implicitly, the attenuation of proinflammatory capacity. In this regard, the first favourable interventions would be the promotion of constant physical activity accompanied by the addition of some natural or artificial factors to dietary intake. Aging is an irreversible process that makes alterations to physical capacity, current and autonomous activities and resistance to stress, and promotes a decrease in physical and cognitive abilities with the onset of pathological aging, especially neurodegenerative disabilities. Thus, by promoting regular physical activity, one can maintain the elderly’s ability to carry out daily activities, to have an independent, safe life, and thus ensure their wellbeing and quality of life. The functional fitness approach represented a very good achievement for improving muscle strength and power, flexibility and mobilisation of lower and upper limb joints, balance, endurance and agility, decreasing the risk of falling and improving the performance of functional movements. In a previous study, reducing training intensity and frequency led to the alteration of functional fitness parameters and negatively modified antioxidant biomarkers [15]. Cognitive aging represents one of the most common manifestations in the elderly, and increasing oxidative stress, as demonstrated by high reactive oxygen metabolites (ROM) levels, remains the biggest problem to be addressed. Furthermore, decreasing antioxidant capacity plays an important role in the pathophysiology of neurodegenerative disorders such as AD, Parkinson’s disease (PD), multiple sclerosis, mild cognitive impairment, etc. [16]. Nutraceutical intervention using natural and artificial compounds represents a new success in healthy aging or delaying the progress of neurodegenerative impairment. Certain foods, rich in flavonoids, proteins, polyphenols or vitamins, have demonstrated beneficial effects in terms of ameliorating blood oxidative stress and inflammation biomarkers [17,18]. 

### 3.1. Instruments/Tools—Determinants of Oxidative Stress, Inflammation Status, Nutraceutical Compounds and Scales of Assessment of Motor and Cognitive Functions on Normal Aging and Aging Pathology

Age-related changes and the correlation with oxidative stress must be approached in the context of the relationship between oxidative stress and inflammatory capacity. From this point of view, there is a need to discuss the most relevant tools for limiting disorders at both the macroscopic and the infrastructural levels. Therefore, the variables required to evaluate and mitigate the impact of oxidative stress and, implicitly, inflammatory capacity, need to be present. Knowledge of pathophysiological mechanisms such as oxidative stress—which influences the installation and progression of motor and cognitive changes for older adults—has led to finding means to limit them. Thus, the most relevant tools in this area are: (1) oxidative stress biomarkers, (2) inflammatory markers, (3) nutraceutical compounds and (4) motor and cognitive scales. All these indicators are used for both normal aging and pathological aging and are presented below in Table 1.

#### 3.1.1. Oxidative Stress Biomarkers/Tools

It is well-known that aging-related oxidative stress and free radicals can cause protein, lipid and DNA molecule oxidation. The process is accelerated by the increase in protein carbonyl molecules through protein oxidation, and large amounts of malondialdehyde and isoprostanes are formed from lipid peroxidation. Thus, increased ROS and NOS concentrations are generated, exceeding the body’s compensatory defence system [2,12]. Nitro-oxidative stress plays an important role in endothelial cell disorders and is implicit in the inflammatory process [14,16]. Oxidative stress is determined by an imbalance between the antioxidant enzyme defence system—represented by SOD, CAT, GPx, GRd—and free radical monitoring through increasing MDA, CP and isoprostanes [2,19,20]. MDA represents lipid peroxidation and oxygen reactive species (ROS) indicators, which are generated in excess when oxidative stress increases [19,21,22]. The increase in oxidative stress reduces mitochondrial activity coupled with oxidative phosphorylation, and implicitly disrupts the balance of redox homeostasis [23]. Cellular aging is the consequence of oxidative stress, which later causes significant tissue damage. The enzymatic antioxidant system (SOD, CAT, GSH, GPx, GR) acts against the oxidative reactive system. The enzymatic complex superoxide dismutase represents the SOD family, which annihilates free radicals, such as superoxide anions. Glutathione is one of the most powerful antioxidants found in all body cells. When it is in small amounts in the body and peroxides increase, pathological mechanisms that induce the inflammatory process are activated. ROS result from molecular oxygen following the processes carried out at the cellular level. The most important endogenous oxidising agents are hydroxyl radicals, hydrogen peroxide and superoxide anions [24]. Therefore, the balances between TOS/TOC and TAS/TAC must be estimated to monitor cellular stress [25]. The 3-NT represents a reactive nitrogen species marker that correlates with the increase in oxidative status [14,21,26]. The accentuation of oxidative stress is correlated with the increase in lipid peroxidation, and the relevant markers in this regard are TBARS and MDA [2,13,22,27]. Protein oxidation represents one of the biggest causes of damage from oxidative stress and can be monitored by determining AOPPs (advanced oxidation protein products), which are an indirect biomarker of accentuated oxidative status [28]. Alzheimer’s disease is one of the most common neurodegenerations and causes progressive dementia, especially on older adults. From a pathophysiological point of view, AD compromises inter-neuronal transmission at the synapse level through amyloid deposit accumulation, intracellular neurofibrillary tangles and pathological synthesis protein generation [15]. Another OS biomarker is phosphatidylcholine hydroperoxide (PCOOH), whose concentration increases by intensifying lipid peroxidation, which disintegrates cell membranes and initiates cell apoptosis [29].

#### 3.1.2. Inflammatory Markers/Tools

Oxidative stress triggers an inflammatory response directly involved in the pathogenesis of diseases. With increasing age, the lipid profile becomes unfavourable and increases the inflammatory capacity, accentuating the imbalance between the antioxidant defence system and prooxidative factors [1,2,13,21,23]. Therefore, the excess production of reactive oxygen species induces mitochondrial dysfunction and maintains a proinflammatory status that becomes chronic through the oxidation of lipids, proteins or DNA. Thus, triggering a cascade of events that initiates cytokine production, such as IL-1, IL-1b, IL-6 and IL-8, which maintains the chronic inflammatory process [10,13,14,15,18,21,30]. Heightened redox potential stimulates the tissue macrophages that produce and release TNF-α into the circulation and upregulates inflammatory mediators, such as intercellular adhesion molecule-3 and BDNF, leading to neuron degeneration [10,21,26]. Their release into the circulation activates blood platelets that cause acute-phase reactants such as CRP and prothrombotic vascular factors (homocysteine, fibrinogen, thrombomodulin, endothelin-1, E/P selectin) [10,14,21,25,26,31].

#### 3.1.3. Nutraceutical Compounds/Tools

The human body has several options for countering the effects of free radicals and oxidative stress based on enzymatic antioxidant molecules, such as SOD, CAT, GPx and GSH, but also nonenzymatic molecules (coenzyme Q10, L arginine) that are endogenous products. Apart from these, there are exogenous antioxidant compounds of animal or plant origin that can be introduced into the body using diet or nutritional supplementation. Next, we discuss the most relevant nutraceutical antioxidants and their protective actions for human health. Astaxanthin and sesamin are carotenoids found in seafoods and fish that have a strong antioxidant effect, preventing muscle fatigue and improving aerobic capacity [23,29]. Dietary nitrate supplementation, such as with beetroot juice, can enhance NO bioavailability, which has favourable effects on peripheral and central haemodynamic responses and metabolic health [26]. Vitamin C supplementation related to iron metabolism change balances pro/antioxidative activity by decreasing proinflammatory cytokine gene expression [2,22,25]. The anti-inflammatory and antioxidant effects of dietary supplementation with capsinoides, along with their improvement to metabolism and decrease in body fat mass, is worth mentioning [32]. The activity of antioxidant enzymes in menopausal women is improved by the addition of phytoestrogen-type flavonoids to the diet. They reduce MDA activity, reactive species and the inflammatory effect of cytokines [13]. Regular cocoa powder that enriches flavonoid consumption has positive effects regarding cardiovascular risk, neurodegeneration and quality of life in older adults [33]. It is also beneficial in neuroprotection; for example, when mitigating mood status and cognitive functions, Cosmos caudatus (CC) turned out to be the high flavonoid integrated in the power supply [34]. Repeated exposure to low temperatures reduces nitro-oxidative stress by improving NO bioavailability and decreasing the activity of inflammatory markers [14]. Diets supplemented with n3-PUFA managed to reduce inflammatory phenomena by reducing the triglycerides concentration and decreasing oxidative stress. Beneficial effects were observed regarding function, muscle mass and anabolic responses. These external antioxidants had positive results for redox homeostasis amelioration [35,36]. Vitamin deficiencies can affect human health, especially cognitive and emotional status, behaviour and personality, causing functional and mental disorders. Apart from the fact that they intervene in numerous metabolic reactions as coenzymes, they play essential roles in maintaining the antioxidant capacity/prooxidative capacity balance by actively participating as supplements in European, Majorcan and Spanish dietary intakes. Vitamin C decreases the inflammatory status by lowering free iron and ferritin levels and decreasing cytokine mRNA expression [25]. Decreasing vitamin D levels can cause muscle impairments such as weakness, pain and diminished adipogenesis [31]. Along with omegA-3, a reduced mitochondrial DNA copy number (mtDNAcn) is considered an oxidative stress biomarker and may improve muscle strength. Omega-3 intake enhances synaptic transmission at the end plate, and it also modulates the contractility of the striated muscle fibre and strength muscle [37,38]. The vitamin B group (vitamin B6, vitamin B12, folate) is associated with a decline in mental disorders and dementia by decreasing HCY trans-sulphuration or re-methylation. Consequently, hyper-homocysteinemia causes DNA metabolism alteration and promotes cognitive impairment and dementia [31,39]. Diet intake supplemented with niacin mitigates the cognitive frailty of dementia [40]. The vitamin E group comprises lipophilic molecules synthesised by vegetal organisms, and the most representative of them is α-tocopherol. Vitamin E inhibits monocyte invasion’s implicit inflammation and depresses oxidative stress by preventing LDL-cholesterol oxidation [41]. Nutritional intake supplemented with vitamin A, vitamin D and beta-carotene mitigates the formation of reactive oxygen species through antioxidant and anti-inflammatory effects [2,25]. Neuroprotective effects associated with aging and with antioxidant potential were signalled by supplementing diets with Laminaria japonica (FST) and desalted Salicornia europaea (PM-EE). Bioactive molecules of FST such as GABA (gamma amino butyric acid) demonstrated an improvement in antioxidant capacity with a protective effect against progressive neurodegeneration. Using PhytoMeal ethanol extract would be safe for seniors with dementia for improving cognitive performance [42,43]. Melatonin is a bioactive molecule with neuroprotective and antioxidant properties. It regulates mitochondrial respiratory complex 1 activity and reduces reactive oxygen species [20]. Molecular hydrogen induced using photo-biomodulation can mitigate cellular oxidative stress and improve mitochondrial functionality, making it a potent and possibly therapeutic antioxidant with neuroprotective effects for Parkinson’s disease [44]. Administration of N acetylcysteine (NAC)—known as a glutathione precursor and an antioxidant agent—is associated with protective effects on brain oxidative stress [45]. Diet intake with benfotiamine, a synthetic precursor of thiamine, can directly participle in multiple metabolic pathways related to oxidative stress or inflammation. Benfotiamine is an antioxidant involved in glucose metabolism, and finally, it decreases advanced glycation end products responsible for the enhancement of reactive oxygen species. Moreover, it increases anti-inflammatory factors in microglia, having an important role in delaying the decline of AD [46]. Ladostigil administration is correlated with delayed dementia progression due to its decreasing effect on microglial activation and reactive oxygen species production. Cerebral atrophy reduction, especially at the medial temporal lobe level, was also mentioned [47]. Polyphenols represent a class of biomolecules that can be natural or biosynthesised compounds. They have acquired strong antioxidant roles and integration into the most diverse food sources due to their ability to reduce free radical oxidation and chelate metal ions, such as iron and copper [10,34]. Symbiotic supplementation such as with kefir improves cognitive decline by alleviating oxidative stress, systemic inflammation and blood cell damage [28]. Lactoferrin, or lactotransferrin (LF), is an iron-building glycoprotein with anti-inflammatory and antioxidative effects. Its presence in the brain is related to aging and neurological disorders. Its participation in protein kinase modulation and tensin homolog pathways justifies the identification of antioxidant and anti-neuroinflammation characteristics [15]. 

We also address nutraceuticals due to their important role in restabilizing digestion and absorption of minerals and vitamins to prevent their deficiency, detoxify cells, inhibit harmful biochemical reactions, facilitate the growth of beneficial microbiota and excrete waste.

The mechanism involving nutraceutical antioxidant effects is based on help from endogenous enzymes when the antioxidant role is less than normal, i.e., the action mechanism used to cure a particular ailment [48] of the human body possesses several antioxidant compounds and enzymes, and their role is to maintain the reactive species’ level of function. Nutraceuticals contain vitamin C, zinc, selenium, vitamin E and enzymes such as glutathione peroxidases and catalases, whose primary job is to scavenge reactive species. This is possible due to nuclear factor erythroid-derived 2-related factor 2 (Nrf2) transcription factor pathway activation. Other action mechanisms from these compounds that have neuroprotective effects are signal transduction cascade modulation and gene expression effects. 

#### 3.1.4. Motor and Cognitive Scales/Tools

The assessment of motor and cognitive functions for healthy elderly people and seniors with neurodegenerative disorders is performed using various scales or tools. The investigation of motor skills is conducted using the following instruments: IPAQ, RAPA and GPAQ, which evaluate physical activity based on questionnaires [17,26,27,49,50]. The physical training score and confidence in balancing activities were evaluated using MET and PASE [18,26,40]. Monitoring posture and balance, both static and dynamic, were quantified using BBS [51]. ADL and IADL are tasks linked to personal care. They refer to activities such as bed mobility, eating, toileting or transfer [40,52]. Power and muscle strength were measured in physical training with weights using 1RM [13]. Apart from the presence of difficulties in physical autonomy, aging speed and pathology are accompanied by cognitive changes influenced by several variables. Instruments used to appreciate degrees of mood, depression or sleep disorders are POMS2, TMD, GDS, S-GDS, PHQ-8, PSQ and PSQI [18,29,30,32,37,50,53]. Disturbances in emotional status have been observed with BDI or BAI scales [51]. For the assessment of mental disorders or cognitive impairments, the following tools have been utilised: MoCA, MMSE and LOTCA [17,28,34,36,39,42,43,46,47,49,49,53]. Neuropsychological profile evaluation has been investigated using NPI, RBANS and NTB [13,14,34]. Instruments used to evaluate quality of life related to independent living, mobility, pain, difficulties, self-care and dietary intake are SF-36-HRQOL, EQ-5D, VAS, NPRS, BDHQ, WHODAS, DHQ, FFQ and Er-Med [16,18,32,33,34,39,40,49,51]. Regarding neurodegenerative disorders such as PD, AD and SM, the tools used for assessing their progression are UPDRS, H&Y, EDSS, SEP-59, MSQOL, ADAS–cog, ADCS–ADL, DAD, CERAD-K and CDR. All these variables evaluate motor and cognitive disabilities as well as the progress of these diseases [15,42,44,45,46,47,54,55]. In addition to those that are specific to the respective neurodegenerative diseases, instruments that evaluate motor status disorders, speech, hearing, vision or logic have been used, such as DS, DSF, DSB, TMT, ATMT, CDT, COWAT, D-FEFS-CWI, RAVLT, CVLT, AVLT, LMT and SDMT diseases [36,39,40,42,47,49,53].

### 3.2. The Impact of Interventions through Physical Activity and Nutraceutical Compounds on Oxidative Stress for Healthy Aging

Concern for the development of autonomous physical activities, motor capacity and muscular abilities in premature aging required finding new intervention strategies to maintain them. Functional autonomy is defined as the capacity to promote daily living activities both at home and in the ambient environment, but also the maintenance of sensory and mental capacities, implicitly promoting quality of life. In order to achieve these desires, it is necessary to find methods of action that mitigate the pathophysiological mechanisms developed in the muscle tissue and in the brain. Aging is often accompanied by disturbances in skeletal muscle contraction and morphology. The characteristic infrastructural signs for the establishment of dysfunction at the skeletal muscle level are protein synthesis alteration, significant mitochondrial-level changes or genomic instability. All these muscular- or nervous-level disturbances are the consequences of the imbalance between the reactive oxygen species production and the body’s defence system in the sense of decreasing antioxidant capacity. Physical activity, regardless of the intensity, frequency, endurance, volume or type, is considered a therapeutic instrument for decreasing oxidative stress [11]. Therefore, one of the strategies for addressing premature aging is the promotion of physical exercise in its most varied forms, starting with daily routine activities and reaching sustained, regular physical training necessary to maintain posture, balance, strength and muscle resistance. In addition to exercise, supplementing the diet with natural or artificial antioxidant exogenous animal or vegetal biomolecules plays a significant role in improving the redox potential and delaying premature aging. Nutraceutical antioxidants, such as vitamins B, C, D, E, polyphenols, omega-3, proteins, flavonoids, oxygen or hydrogen therapy, showed beneficial effects in the health of older adults. 

Below, we present Table 2, showing previous studies describing aspects of physical activity and nutraceuticals and oxidative stress according to different types of physical activity and dietary intake.

Analysis of the references provided demonstrates the effects of endurance training performed on the treadmill for 12 weeks, with 3 sessions/week, or physical tasks for 4 h/day with a cycle ergometer for 4 weeks. In addition, near-physical and mental tasks were applied for 4 h/day in sets of 30 min, using an advanced trail-making test. In both cases, a nutraceutical antioxidant diet with astaxanthin (AX) was administrated, represented by red, powerful, oceanic carotenoids found in algae, shellfish and fish, or AX with sesamin, i.e., the oleaginous seed found in sesame. After the two interventions in both sexes, through sustained physical activity and a supplemented diet, beneficial effects regarding the lipid profile through increased fat oxidation, decreased carbohydrate oxidation and an anti-fatigue effect through lowered plasma phosphatidylcholine hydroperoxide were observed during physical and mental tasks. By applying the trail-making test for 4 weeks, improvements were observed in the emotional status and quality of sleep in older adults [23,29]. Moreover, some other authors have shown the effect of the physical intervention, quantified by cycling 10 M with a pedalling cadence of 50 rpm with moderate or intense effort. As a supplemented diet, 70 mL × 2/day of beetroot juice (dietary nitrate NO_3_^−^) was used for 7 days. The results did not show a favourable influence on haemodynamic and cardiac parameters, but changes in inflammatory and oxidative markers such as E/P selectin, thrombomodulin and ICAM-3 were observed. IL-6, 3-NT and glycaemic levels were quantified in a positive sense using HOMA-IR [26]. In other writings, the authors evaluated the results obtained after applying physical training and dietary intake vitamins of type A, C, E, tocopherol and beta-carotene, or trace elements such as Zn and Se. Physical exercises were carried out for 6 weeks, 3 times/week for 60 min, either using cycle ergometer sessions or gyrokinesis, Nordic walking and stabilisation training. Other options were physical fitness or intermediate and active exercise training. Vitamin therapy was administrated using tablets (1000 mg/day, 6 weeks) or nutraceutical food with vitamins. The studies showed that anthropometric index, muscle mass, fat mass and cardiorespiratory parameters underwent favourable changes, such as a decreased BMI and increased muscle mass. Beneficial effects were observed in oxidative stress and inflammatory marker levels. Improved antioxidative/prooxidative balance and anti-inflammatory/proinflammatory capacity were monitored using SOD, CAT, GRx, GRd, 3-NT, MDA, TBARS, protein carbonyls or isoprostanes, and IL-1, Il-6, IL-10, CRP, tubulin beta class 1 and CCL2 (chemokine ligand 2). In conclusion, a decrease of about 10% in oxidative stress and enhanced antioxidant defence, changes in lipid profile and glycaemic level, decreased IL-6, increased Il-1 and increased oxidative metabolism capacity after regular physical activity and supplementation with vitamins and oligomolecules were noticed [2,22,25]. Physical activity of various degrees, from low to moderate or vigorous, monitored using an activity meter together with the consumption of 9 mg/day of capsinoids (CH-19 sweet) for 12 weeks, was determined in terms of body composition, decreased waist circumference, visceral fat index and body fat percentage through lipid oxidation, augmenting the physiological consequences of ingestion of capsinoids. The tools that quantified the effects were FAS, METs and parameters of mood status such as POMS2 and TMD. Other effects of interventions were increased energy expenditure in sedentary participants, homeostatic capacity regulation, with a lowered occurrence of chilly sensation in older adults, and enhanced oxidative phosphorylation in striated muscles. Capsinoid consumption stimulated physical activity in the sedentary elderly of the CP group [32]. Another nutraceutical option alongside resistance and aerobic training was supplementation with 100 mg/day of isoflavone for 10 weeks. This study evaluated the effect of an isoflavone diet mixed with physical exercises and investigated the lipid profile, oxidative stress and inflammatory markers in postmenopausal women. The physical training comprised mixed aerobic (carried out with a treadmill) and resistance activity over 30 sessions for 50 min/session, 3 times/week, mediated using the 1RM test with eccentric and concentric phases. Beneficial results were registered, with increased IL-8 levels being an important angiogenic agent, and lowered total cholesterol, which is associated with the anti-inflammatory status [13]. By applying whole-body cryotherapy, over 24 exposures for 3 min, 7 times/week for 7 days, and endurance training (long-distance runners), another author demonstrated no change in the 3 nitro-tyrosine level and no change in nitro-oxidative stress in male seniors. The increasing iNOS concentration and implicit enhancing bioavailability of nitric oxide demonstrated that WBC has a beneficial effect on vascular vasodilatation and important antioxidant and anti-inflammatory results, such as stimulating lipid oxidation through the metabolic effect of IL-6 and decreasing IL-6 levels in the RUN (long-distance runners) group. Nitro-oxidative stress and inflammation markers did not change after exposure to WBC [14]. Aging is accompanied by progressive loss of function and mass skeletal muscle-related frailty. Sarcopenia can cause mobility disorders, risk of falls, lack of independence and alteration to metabolic health in the elderly through mitochondrial disturbances, insulin resistance and oxidative stress. Another study demonstrated that the association between acute or resistance training with n3-PUFA supplementation for six months determined a modest increase in skeletal muscle strength by decreasing inflammatory effects and reducing oxidative stress at the mitochondrial level. Improving the contraction energy level after n3-PUFA during exercise training and decreasing the rate of protein synthesis attenuated functional disorders, sarcopenia and chronic inflammation in skeletal muscle. The addition of omega-3 to diets improves muscle strength in elderly adults with normal glycaemia [35,38]. Another paper, namely the PHYSMED project, observed an association between physical fitness of different degrees and blood biomarker levels associated with inflammatory status or oxidative stress. Therefore, the lower physical fitness group obtained decreasing triglyceride and vitamin B12 concentrations and increased values outside the normal range for homocysteine, creatinine and the lipid profile. In the vast majority of elderly Spanish people, vitamin D(OH) values were low. In the three groups, low-, medium- and high-fitness dietary intake was seen in European, Spanish or Majorcan food. Regular physical activity and a diet associated with a healthy lifestyle and delayed premature aging were found for the high-fitness group [31]. A study carried out based on the association between increased acrylamide consumption and physical activity showed a poor capacity for effort and even negative effects regarding muscle mass quality, implicitly accentuating sarcopenia and obesity in the elderly. Monitoring was performed using walking training, chair stands repeated five times, FFQ and PASE [18]. A higher dietary intake of flavonoids (cocoa beverage powder, once/day, 12 weeks) showed positive effects in muscle strength and resistance, increasing exercise capacity, and reducing anti-inflammatory effects and oxidative stress. Physical performance, cognitive profile and quality of life reflected through SMI and EQ-5 were improved in older adults [33]. Nutraceutical dietary supplementation with L-glutamine (Gln) for 30 days, demonstrated in another study, increased the physical capacity regarding knee muscle strength/power in elderly women using endurance and strength training, 60–70 min/session, 3 times/week, monitored using an isokinetic test. Glycaemic-level regulation using lower insulin concentrations and oxidative stress amelioration with increased oxidised glutathione (GSH, GSSG) and lowered thiobarbituric acid-reactive substances were also mentioned [27]. In a large study with 391 elderly participants (non-Hispanic white, Black and Hispanic populations) and with habits such as cigarette smoking and alcohol consumption, physical activity testing was carried out, graded in effort intensities. The diet was supplemented with vitamin D and omega-3 (VITAL-DEP program). The mitochondrial DNA copy number is a biomarker associated with premature aging, oxidative stress and behavioural habits. The monitoring of the effect of low, intermediate and vigorous physical activity and the supplemented diet was carried out using METs, PHQ-8 and mtDNAcn. A lowered mtDNA copy number for smoking participants was registered as a negative impact for elderly Black people [37]. It is known that exposure to ozone causes endothelial and cardiovascular effects and is correlated with inflammation and an increase in reactive oxygen species due to the absence of the glutathione S-transferase Mu1 (GSMT1) gene. However, controlled exposures to ozone with concentrations increasing from 0 ppb (parts per billion) to 120 ppb, together with moderate exercise training in elderly adults, did not determine obvious inflammatory effects, but increases in CRP 8-isoprostane and Il-6 and endothelin-1 were recorded in another study. A decrease in 3-NT and nitrosative stress levels was reported as a positive effect [21]. It is not clearly stipulated that dietary intake supplemented with protein can improve muscle injuries and fatigue induced by prolonged walking exercises in elderly adults. The study carried out on the target and placebo groups after walking training for three consecutive days showed comparable increases in creatine kinase following intense muscular effort, and protein supplementation in the study group did not demonstrate faster elimination of muscle pain or fatigue compared to the control group. Inflammatory markers such as CRP, IL-6 and Il-10 had comparable values in both groups because the muscle protein synthesis response to anabolic stimuli was diminished [30].

Endurance training performed on a treadmill for 12 weeks with 3 sessions/week or physical tasks for 4 h/day with a cycle ergometer for 4 weeks is recommended. Mental tasks for 4 h/day with sets of 30 min, using an advanced trail-making test, have also been applied. The administration of a nutraceutical antioxidant diet with astaxanthin (AX), represented by powerful, red oceanic carotenoids found in algae, shellfish and fish, or AX with sesamin, i.e., oleaginous seed found in sesame, is also recommended.

### 3.3. The Influence of Physical Activity on Oxidative Stress for Healthy Older Adults

Aging is traditionally accompanied by functional autonomy restriction, skeletal muscles’ capacity decline, frailty, risk of falls and, often, cognitive disorders. Low muscle strength found in older adults, legitimised by evidence of a slowed-down skeletal muscle response to anabolic agents, accentuates motor disabilities and the risk of mortality. Therefore, the intervention of strategies that address both the maintenance and recovery of functional deficiencies as well as the slowing down of cognitive decline is required. The concern for promoting a healthy lifestyle and maintaining quality of life must be made a priority in order to prevent or delay premature aging. Physical activity (PA) in its various forms, such as physical exercises, resistance or endurance training, or both, represents an essential direction in promoting healthy aging at the motor level and in the cognitive domain. One aspect of age progression is damage caused to the balance between the antioxidant defence system and reactive oxygen species. Oxidative stress initiates proinflammatory factors, which causes muscle frailty and motor function decline and affects mental health. The main results regarding different PA program recommendations that could influence the oxidative stress are presented in Table 3. 

Two references discussed the association between physical activity of different degrees of intensity (low, moderate and vigorous) and physical fitness (PF) associated with muscular and cardiorespiratory endurance, muscular strength, flexibility and body composition. Within the wide PREDIMED program with 6874 participants, the results of the interventions were monitored using MMSE and TMT, showing an improvement in neurocognitive parameters linked to physical fitness. Enhanced scores for verbal and phonemic verbal fluency were quantified using COWAT, CDT and DS, and increased quality of life using SF36-HRQOL. Physical fitness was associated with better cognitive activity [49]. The frailty index decreased and motor function, mental and emotional status and sleep quality improved, as measured using IPAQ-SF, PSQOI and MMSE. PF levels together with PA favourably contribute to motor and nonmotor skills and functions [50]. Moderate- or high-level PA demonstrated a greater increase in plasma α- and γ-tocopherol and a decrease in oxidative stress and homocysteine levels than in senior individuals who performed low-intensity physical activities. Exercise training is associated with cognitive function; thus, markers such as vitamin E, homocysteine level and free radical species can be used as predictive factors for assessing mental health in elderly adults. Monitoring was evaluated using GPAQ, LOTCA, METs and MMSE [17]. Regular low- or moderate-intensity physical activity stimulates antioxidant and anti-inflammatory capacities, as evidenced by increased SOD or GPx and decreased MDA, Il-6 and TNF-α levels. Moreover, 12 weeks of Taekwondo martial art training for 60 min, 4 times/week, showed an improvement in motor function and agility motions, and ameliorated depression, anxiety and sleep disorders in terms of antioxidative and anti-inflammatory responses in postmenopausal women [19]. Multimodal exercise training (back-scratch, chair sit-to-stand or grip strength), such as functional fitness for 24 weeks, 1 hour/session, twice per week, improved flexibility motions, muscle strength, dynamic balance and other biomechanical parameters. Regarding blood biomarker levels, the study measured decreased reactive oxygen metabolites, prooxidant activity and hormone stress concentrations (cortisol). Functional fitness by performing daily tasks increased upper and lower limb flexibility, muscle strength, cognitive function amelioration, and decreased body fat percentage and prooxidative status. Increased quality of life and mental health was observed using EuroQol-5 [16]. Another study assessed the involvement of daily physical activity from moderate-to-vigorous levels for 8 weeks on postmenopausal women using an axial accelerometer. After exercise training, the observed findings were an increased step count, brain-derived neurotrophic factor and serotonin levels. These results demonstrated decreased oxidative stress quantified using biological antioxidant potential (BAP) and prevented depression, measured using GDS [52]. Advancing age is associated with suffering of the striated muscle mass and function regarding oxidative damage biomarkers. In another study, the author demonstrated the influence of resistance training for 6 weeks, twice/week, using 3 sets with 10–12 repetitions, on skeletal muscle redox potential. Oxidative stress markers such as 4-hydroxynonenal, heat-shock proteins (HSP60) and protein carbonyls (PC), using CAT, SOD, GDS and GAPDH, were investigated. After resistance training carried out on many levels, positive results were identified from muscle biopsies, mRNA and various endogenous antioxidants [11]. The risk of cognitive disorders in elderly adults is a serious mental health problem. Another reference discussed the contribution of a game-like dual-task exercise, named “synapsology” (SYNAP), which improved physical and cognitive responses in older adults. Physical activity and cognitive profiling were carried out over 8 weeks, 60 min/session, twice a week, and investigated using timed up-and-go (TUG), the five-time sit-to-stand movement test (5XST), the Trail-Making Peg Test (TMPT), reactive oxidative metabolites and brain-derived neurotrophic factor (BNDF). Beneficial results have been observed after using the SYNAP program to assess cognitive status, daily functional abilities and healthy life [54]. A hydrotherapy program performed over 15 sessions stimulated the antioxidant profile by upregulating glutathione reductase, peroxidase activity and superoxide dismutase enzymes, and decreasing reactive oxygen species in the body. Thus, regular exercise training prevents disorders caused by premature aging [24]. A sedentary lifestyle and behavioural factors such as smoking and coffee and alcoholic beverage consumption on older women is associated with oxidative stress, and progressed aging was demonstrated in another study. OS was measured using MDA, SOD, GPx and the SOD/GPx ratio. A stress score ranging from 0 to 7 was associated with marker modification. A stress score of ≤4 and physical activity of <30 min/day in sedentary Mexican women were associated with increased oxidative stress [1]. In the same sense, another author discussed the importance of moderate physical activity associated with oxidative defence capacity on free radicals and progressed aging. Increased SOD/GPx ratio, SOD, CAT and GPx levels in blood compared with decreased MDA, carbonyl protein and isoprostane concentrations were noted in older adults who participated in moderate exercise programs versus the sedentary elderly. In conclusion, it was demonstrated that physical activity provides an antioxidant-protective effect by decreasing free radicals and proinflammatory markers [12].

### 3.4. The Impact of Recovery Using Physical Activity and Nutraceutical Compounds on Oxidative Stress for Neurodegenerative Diseases

Neurodegeneration is the result of cerebral metabolism disorders, glial system-level changes, as well as neurotransmitter communication alterations in synaptic networks and abnormalities at the blood–brain barrier, such as endothelial cell lesions. Oxidative stress occurs in neurodegeneration disorders such as Alzheimer’s disease (AD), Parkinson’s disease (PD), amyotrophic lateral sclerosis (ALS) and multiple sclerosis (MS). In AD, the neuropathological mechanism is described as an amyloid protein accumulation outside the cell and the presence of neurofibrillary tangles composed of hyper-phosphorylated “tau proteins”. Thus, synaptic connection loss occurs in selective brain regions. Along with OS, mitochondrial function is also affected by an increase in free radicals that leads to nerve cell apoptosis. PD is defined as a loss of melanin-pigmented nigral neurons with dopamine depletion in the basal ganglia and the presence of Lewy bodies. By increasing oxidative stress, it accentuates the destruction of the substantia nigra and the acceleration of cell death. In ALS, increased accumulations of carbonyl protein, nitro-tyrosine and superoxide radicals lead to certain irreversible cellular alterations. In SM, intensive lipid peroxidation is activated, which leads to demyelination, which causes severe neural destruction. Decreasing SOD levels and enhancing TBARS and nitrite concentrations contribute to the amplification of oxidative stress, proinflammatory capacity and damage to nervous cells. Therefore, the need to improve morphological and functional neuron decline and delaying pathophysiological neurodegeneration mechanisms has prompted the intervention of neurorehabilitation strategies and methods of prolonging the evolution of neurodegenerative diseases. Thus, the combined promotion of physical activity in various forms and intensities and nutraceutical compound diet participation has demonstrated favourable effects in limiting the oxidative stress and inflammatory phenomena developed in neurodegenerative impairments. In Table 4, we present the relationship between different types of PA, dietary intake and oxidative stress.

Usual and enhanced physical activity, such as aerobic exercise training using a treadmill for 26 weeks, 150 min/week, in 3 sessions/week, and dietary intake with phospholipids and PUFAs, were performed on 23 older adults with familial and genetic risks of AD. Blood systemic biomarker levels involved in cognitive function (memory, learning), such as myokine cathepsin B (CTSB), BDNF, lipid profile and klotho, were adjusted after physical training, which showed positive mental wellbeing and brain function effects [36]. In another study, the influence of physical fitness, ADL and IADL with a handgrip and additional low-niacin dietary intake on mental health, depression or emotional disorders was assessed. A total of 815 participants with frailty and pre-frailty cognitive status took part in a LRGS-TUA study. They were monitored using nonmotor scales such as GDS, MMSE, MoCA, RAVLT, digit span, WHODAS, DHQ and SOD, MDA, BDNF and DNA damage. The telomerase level results demonstrated beneficial effects on oxidative stress amelioration and cognitive frailty identification in elderly people with mild impairments or dementia [40]. Aerobic training with low intensity carried out in an aquatic environment for 12 weeks, 2 times/week, 45 min/session, was credited with decreasing depression and anxiety scores, functionally independently, and the implicit decrease in reactive oxygen species. The quantification was carried out using the Borg scale, BAI, BDI, TUG, GSH, SOD NO and protein carbonyl levels [51]. Dietary intake of a functional food such as Laminaria japonica (FST) from seaweed—which has a six-week antioxidant effect—together with physical fitness training provided protective action against progressive neurodegeneration caused by free radicals. The results measured after the interventions were improved physical autonomy, cognitive functions and antioxidant activity, indicated by increased SOD, GSR and GPx levels, and the capping of oxidative stress markers such as TBARS, 8-oxoDG and BDNF. Mental health evaluation monitoring was used, including the MMSE, the numerical memory test, Raven’s test, Flanker test and the iconic memory test, with positive results [43]. High-intensity interval training performed using a cycle ergometer, weights, a dynamometer and elastic bands for 12 weeks, with 1 session/week, demonstrated a reduction in fatigue and spasticity, and an improvement in muscle strengthening, resistance and aerobic performance, implicitly increasing the quality of life. The results obtained using QOL, SEP-59 and MSQOL-54 demonstrated the possibility of carrying out autonomous activities and improving living conditions [55]. AD dementia is associated with both autonomous functional decline and that of cognitive status. Daily regularly exercise in combination with desalted *Salicornia europaea* L. (SE), a species of halophytic plant, was shown to improve frontal executive functions, such as oxidative stress amelioration, proinflammatory biomolecule decrease and neuroprotective pathway stimulation. The program was carried out for 12 weeks by administering PhytoMeal (SE) ethanol extract, 600 mg/day, with physical training. Notable performances were observed in cognitive tasks assessing perception, attention, working memory and language using ADAS–cog, CERAD-K, K-CWST and S-GDS. Beneficial results were registered regarding quality of life in older adults with AD, such as an increased ability to perform daily living activities.

### 3.5. The Effects of Recovery Using Nutraceutical (Antioxidant) Biomolecules on Oxidative Stress for Neurodegenerative Diseases

Neurodegenerative disorders are associated with oxidative stress and mitochondrial dysfunction via protein aggregates, which compromises the activity of mitochondrial enzyme complex I, thus stimulating free radical production. Therefore, the use of natural or artificial agents with antioxidant action capable of limiting and delaying motor and cognitive disorders within the neurodegenerative process is required. Daily supplementation with 50 mg of melatonin for 3 months decreased reactive oxygen species through increasing mitochondrial complex I and the respiratory ratio, catalase and superoxide dismutase, and lowering glutathione, malondialdehyde and carbonyl protein levels in older adults with Parkinson’s disease. The beneficial effects of melatonin refer to the reduction in mitochondrial dysfunctions from oxidative stress and the proinflammatory effects [20]. The relationships between antioxidants and oxidative stress in neurodegenerative diseases are presented in Table 5.

A strong antioxidant effect on pathophysiology neurodegeneration has been mentioned for molecular hydrogen (H_2_). Molecular hydrogen intake was carried out through the daily ingestion of H_2_—water associated with photo-biomodulation (PBM)—for 2 weeks, or inhalation of 6.5 vol% H_2_ gas in 2 L/min for 16 weeks, twice/day, and mixed air was obtained using electrolysis in patients with PD. The effects of molecular hydrogen administration were decreased reactive oxygen species, stimulated mitochondrial activity with enhanced ATP levels and mitigated cognitive function deterioration [44,54]. Another author discussed the intravenous administration of N-acetylcysteine (NAC), a powerful antioxidant that stimulates brain glutathione levels, 6000 mg/day for 28 days, in elderly people with PD. Oxidative stress indicators such as MDA or 4-hydroxynonenal (4-HNE) remained unchanged, while antioxidant biomarkers of the same type reduced the oxidised ratio (GSH/GSSG), and catalase increased in the concentrations [45]. Another antioxidant used as a supplement was benfotiamine, a synthetic thiamine derivative (pharmacokinetic marker belonging to the vitamin B group) administered orally for 12 months, 300 mg twice/day. Spectacular results were obtained in terms of cognitive and functional status, demonstrated using ADAS–cog, CDR, MMSE, NPI and ADCS–ADL scales. Therefore, ADAS–cog and CDR scores decreased by 43% and 77%, respectively, after benfotiamine administration with amelioration of cognitive abilities, such as memory, praxis, language, attention, orientation, judgement and verbal memory. In this sense, improvements were observed in daily activities carried out independently and the ability to make one’s own decisions and judgments. Decreased oxidative stress and proinflammatory activity were demonstrated by lowered blood biomarkers such as advanced glycation end-product levels [46]. Dietary intake of vitamins from group B, especially vitamin B12 and folate, was used in an EMCOA study, showing decreased cognitive decline, especially with better cognitive reserves and MDA, HCY, 8-OHdG and 8-isoPG2α oxidative stress marker types. The results were monitored using mitigated MMSE, MoCA, AVLT, DSF, DSB and SDMT scores [39]. Mild cognitive disorders represent a heterogeneous syndrome defined by a decline in memory performance, with this being a transitional line between normal aging and dementia in terms of neurodegenerative diseases. The addition of Cosmos caudatus, a vegetable of the Cosmos-type annual plant, turned out to have beneficial effects on mood status amelioration and global cognitive status improvement after 12 weeks with 250 mg twice daily. Lowered lipid peroxidation reduced oxidative stress, proven by assessing MDA, BNDF, SOD, iNOS, GSH and COX-2 levels. The scales used for measuring mild cognitive impairment were MMSE, digit span, RAVLT and POMS [53]. Another author discussed alleviating neurodegeneration, especially in medial temporal lobe atrophy after low-dose ladostigil administration at about 10 mg/day for three years. The quantification of favourable responses was performed using MMSE, WMS-RC CDR, DAD, RAVLT and GDS scales [47]. Dietary intake with polyphenols combined with the MindFoods (cognitive training) program was proven to decrease reactive oxygen species and proinflammatory cytokines after supplementation for 12 weeks in older adults with AD. Improved neurobiological health and enhanced anti-inflammatory cytokine levels were demonstrated using MMSE, TMT, GDS and RBANS scores, and TBARS and mRNA markers [34]. The antioxidant effects of probiotics such as kefir using nutraceutical supplementation for 90 days at 2 mL/kg/day were proposed in another study that measured systemic oxidative stress, proinflammatory and anti-inflammatory cytokines, NO bioavailability and cognitive function or capping metabolic disorder improvements. The assessment was performed using MMSE, immediate memory delay, TMT-A, the clock-drawing test, the delayed memory test and the Boston test, whose scores improved in terms of visual–spatial function, language abstraction, memory and conceptualisation skills. Oxidative stress (ROS, AOPP, PARP-1), systemic inflammation (IL-8/IL-10 ratio, IL-12/Il-10 ratio, Il-6, TNF-α, IL-1b) and metabolic cellular damage were modulated using the probiotic antioxidant intervention [28]. Another study proved the antioxidant and anti-inflammatory role of lactoferrin, a multifunctional glycoprotein iron fastener, after administration for 3 months at 250 mg/day. The favourable intervention of lactoferrin using the modulation protein kinase B/phosphatase and tensin homolog (PTEN) pathway and acetylcholine (Ach) and serotonin (5-HT) serum levels was demonstrated. Decreased MDA, IL-6, heat-shock proteins, cholesterol and tau proteins, and increased SOD, GSH and IL-10 levels validated the antioxidative and anti-inflammatory effects of lactoferrin. The positive cognitive function results were monitored using MMSE, ADAS–cog 11 and CDR. Thus, lactoferrin administration seems to be a protective intervention, modulating oxidant and proinflammatory pathological cascades and cognitive decline [15].

## 4. Discussion

Biological aging is characterised by decreasing functional and cognitive abilities, being much more pronounced in elderly adults’ motor impairments and cognitive decline. Thus, functional autonomy loss as well as a decrease in mental and behavioural progress accelerate the onset of motor decline and dementia. Hence, these aspects cause great concern for the health and care systems. Although these disorders have multiple causes, the most publicised hypothesis in this regard is that related to oxidative stress. Disturbing the balance between the body’s antioxidant defence power and the production of free radicals leads to mitochondrial dysfunctions and cellular metabolism damage. Reactive oxygen species and proinflammatory cytokines are thus generated, which promote lipid peroxidation, hyperphosphorylation or oxidised proteins, and DNA damage. Synthesis of the relationship between oxidative stress markers and normal aging, physical activity, pathological aging and nutraceutical compounds is presented in Figure 1. 

These processes create abnormal amyloid beta–peptide aggregations and intracellular neurofibrillary tangles, disturbing synaptic networks with changes in acetylcholine, dopamine and serotonin levels, which generates neuroinflammation and neurodegeneration. In our study, we performed a review of the research findings in the literature regarding how different types of physical activity and supplementation with natural and artificial nutraceutical compounds could delay the aging process and the development of neurodegenerative disorders. Physical activity, starting from ADL and IADL, and continuing with walking training, resistance or endurance exercises, physical fitness, multimodal exercises or physical tasks, managed to maintain and improve functional autonomy and delay cognitive decline. Applying moderate-intensity training for six or twelve weeks, such as habitual activities, Nordic walking, gyrokinesis, 400 m walking or chair stands repeated five times, showed significant improvements in VO_2_ max, systolic and diastolic blood pressure, BMI, muscle mass or fat mass, physical frailty, mental health, emotional status and sleep disorders. MMSE, IPAQ-SF, PSQOI, PASE, MoCA WHODAS and GDS scores demonstrated beneficial results in healthy elderly adults and seniors with neurodegeneration for both genders [1,18,25,33,40,50,52]. Promoting low, moderate or vigorous physical activity (LMVPA) contributed to improving the gait pattern, strength, gait, physical mobility, skeletal muscle strengthening, body fat percentage, BMI and the visceral fat index, including oxidised stress parameters.

In the lipidic modulation profile, decreased HCY, 3-NT, MDA, IL-6, CRP and mtDNA copy number, or increased SOD, CAT, GRd and GPx, resulted after the physical training intervention at different intensities applied for 6–8 weeks for 75 min/week, 150 min/week or 5.5–30 MET hours/week. Increased functional autonomy, quality of life and enhanced cognitive capacities, including neurocognitive parameters, were monitored using MMSE, COWAT, DS, CDT, TMT, METS, LOTCA and GPAQ scores [12,14,17,21,22,32,35,37,38,49]. Endurance or resistance training were another form of physical activity, using devices such as a treadmill, cycle ergometer or handgrip dynamometer, being applied 6–12 weeks for 60 min/session, 3 times/week. Improved antioxidant or anti-inflammatory status, cardiovascular, respiratory and immunological parameters, and implicitly increased motor and cognitive functions, both for normal and pathological aging, were measured. The tools that evidenced beneficial effects for both healthy older adults and seniors with neurodegenerative impairments were MMSE, FFT, TUG, 5xST, POMS2, COWAT, DS, TMT, CDT, LOTCA, PSQOI and GDS, with influence on motor functions, neurocognitive capacity and quality of life [10,13,19,23,24,26,27,30,31,36,50,51,52,55]. Physical fitness is another way to evaluate oxidative stress reduction and improved cognitive abilities by measuring the ameliorating effects on neuromuscular integrity, sleep disorders, emotional and mental motor status and cognitive scales, such as a lowered risk of frailty index. Positive results were demonstrated for both genders and for healthy or neuropathological elderly adults using BDHQ, MMSE, IPAQ-SF, PSQOI, PASE, RAVLT, DS, MOS-SS, GDS, numerical, the iconic memory test, TMT or Raven’s test [2,40,43,50]. Multimodal exercises determined enhanced flexibility in upper and lower limbs, and improvements in dynamic balance, endurance and muscle strength, evidenced by alleviated oxidative stress and proinflammatory capacity [16]. A special approach was the implementation of physical tasks using a visual display, a game-like dual-task or ergometer task, with significant improvements in attention, working memory, spoken language and motor functions [29,42,56]. Another direction for promoting normal aging without unfavourable events and mitigating neurodegenerative disorders in older adults is the promotion of nutraceutical dietary supplements with natural and artificial antioxidants administered using diet or medication. The antioxidant power of vitamins A, C, D, E, complex B, folate and their derivatives were recognised. Using these interventions allowed to obtain increased antioxidant agents, demonstrated by the activation of potential redox enzymes such as SOD, CAT, GSH, GRd, GPx and TOS/TOC and TAS/TAC, which attenuated cellular and mitochondrial disruptions, and showed anti-inflammatory effects. The beneficial action of vitamins is expressed by the modulation of pathophysiological biomolecule synthesis pathways that act in free radicals and proinflammatory markers (IL-6, Il-8, IL-1, CRP, TNFα, BDNF). Antioxidant and anti-inflammatory effects improve functional capacity and mental health in elderly adults of both genders [2,22,25]. Along with vitamins D and E, other nutrients intervene in the same sense, such as omega-3 and n3-PUFA, which strengthen the antioxidant effect, modulate lipid and carbohydrate profiles, increase phospholipids and intracellular and mitochondrial ATP levels, and decrease concentrations of inflammatory substances, such as BDNF and Klotho proteins. Enhanced motor and cognitive function were demonstrated using these interventions in healthy older adults and elderly patients with neurodegenerative diseases [35,36,37,38]. Other nutraceuticals were supplemented using dietary proteins and N-acetyl cysteine. The results showed favourable effects on antioxidant/oxidant agent balance by using CAT, GSH/GSSG, SOD/GSH, MDA, 4-HNE, protein carbonylation and scores of scales such as IPAQ, NPRS, UPDRS, H&Y, BDI, BAI and BBS. They improved physical or neuromotor functions for normal aging and ameliorated depression, anxiety and sleep disorders in seniors of both sexes with neurodegeneration impairments [27,30,45,51]. European, Spanish or Majorcan foods have made significant contributions to functional capacity and reactive oxygen species’ decrease. The results were demonstrated using TDW, FFM, LET, QoL SEP-59 and MSQOL [14,31,55]. Supplementation with polyphenols, flavonoids, capsinoids, Laminaria japonica, Cosmos caudatus or acrylamide showed the same aspects regarding oxidative stress and favourable results regarding motor abilities and mental functions. PASE, SMI, EQ-5D, FFQ, VAS, TMD, POMS2, LBM, MMSE, CERAD-K, K-CWST and GDS demonstrated progress in terms of motor and cognitive autonomy effects [13,18,32,33,42,53]. Nitrate dietary supplementation had no favourable influences on neither cardiac and haemodynamic parameters, nor oxidative stress. After taking small doses of ladostigil, neurodegeneration due to dementia was mitigated. Quantification was carried out using MMSE, CDR, WMS-RC, NTB, DAD, RAVLT and GDS, which estimated medial temporal lobe atrophy and showed favourable evolution in mild cognitive impairment [47]. The proposal for supplementation with ozone or molecular hydrogen did not provide effective results regarding oxidative stress, but the administration of molecular hydrogen together with photo-biomodulation stimulated mitochondrial activity with increased ATP levels, associated with alleviating cognitive disorders [21,44,54]. The presence of probiotics such as kefir demonstrated memory improvement, language abstraction and conceptualisation skills and decreased reactive oxygen species and proinflammatory biomarkers. A beneficial antioxidant effect was proven in the progress reflected using cognitive scales such as MMSE, immediate and delayed memory tests, TMT-A and the clock-drawing test [28]. The addition of melatonin for three months at 25 mg/day proved beneficial in reducing oxidative stress biomarkers when analysing CAT, mitochondrial complex I activity and the mitochondrial respiratory control ratio [20]. Decreased free radicals and proinflammatory agents demonstrated by assessing afferent blood biomarker levels and cognitive scales such as MMSE, CDR and ADAS–cog 11, showed mental function alleviation in elderly patients with AD dementia.

## 5. Conclusions

The aging process reduces physical capacities, alters the cognitive status and is related to oxidative stress, which is defined as an alteration in the balance between oxidant and antioxidant biomolecules. This is because ROS leads to genetic, molecular, cellular, tissue and systemic changes.

In the search for healthy aging and a delayed pathological neurodegenerative evolution process in elderly people, mechanisms must be promoted to adjust the imbalance between the excessive production of ROS and the decrease in the defensive enzyme system and, implicitly, proinflammatory capacity attenuation. Therefore, the promotion of constant physical activity accompanied by the addition of natural or artificial factors to dietary intake is the first favourable intervention in this regard.

Protein oxidation represents one of the biggest damages caused by oxidative stress and can be monitored by determining AOPPs (advanced oxidation protein products), being an indirect biomarker for an accentuated oxidative status.

Regular physical activity and supplementation with vitamins and oligomolecules elicit decreased IL-6 and increased IL-1 oxidative metabolism capacity. In conclusion, this study demonstrated that physical activity provides an antioxidant-protective effect by decreasing free radicals and proinflammatory markers.

## Figures and Tables

**Figure 1 antioxidants-12-01008-f001:**
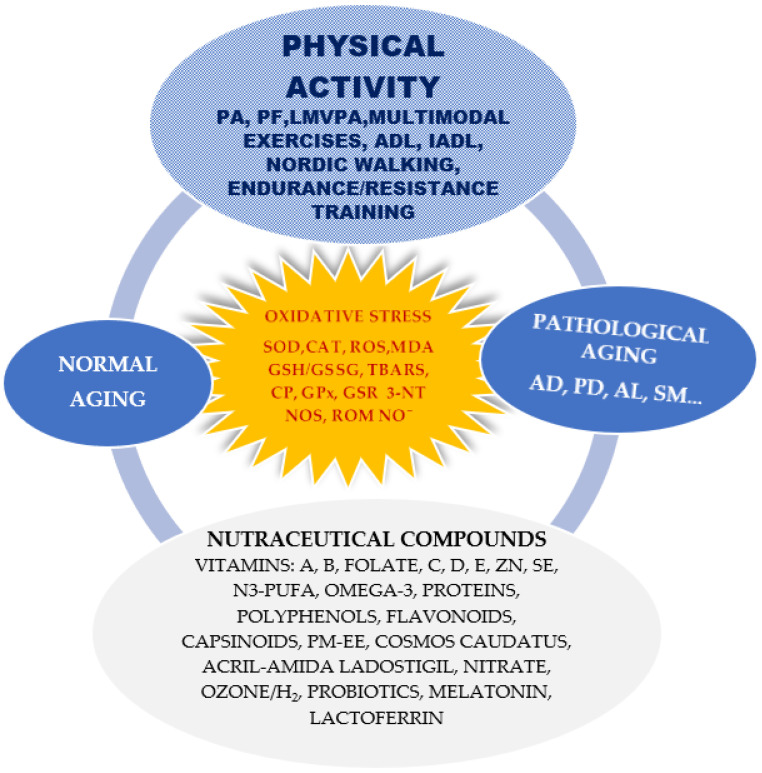
Relationship between oxidative stress, physical activity and nutraceutical compounds in the aging process. Legend: PA—physical activity; PF—physical fitness; ADL—activities of daily living; IADL—instrumental activities of daily living; SOD—superoxide dismutase; CAT—catalase; ROS—reactive oxygen species; MDA—malondialdehyde; GSH/GSSG—reduced/oxidised glutathione; TBARS—thiobarbituric acid-reactive substances; CP—protein carbonyl; GPx—glutathione peroxidase; GSR—glutathione reductase; 3-NT—3 nitro-tyrosine; NOS—nitric oxide synthase; ROM—range of motion; AD—Alzheimer’s diseases; PD—Parkinson’s disease; ALS—amyotrophic lateral sclerosis; MS—multiple sclerosis; N3-PUFA—omega-3-polyunsaturated fatty acids; PM-EE—PhytoMeal ethanol extract.

**Table 1 antioxidants-12-01008-t001:** Oxidative stress, inflammatory markers, nutraceuticals and motor/cognitive scales/tools/instruments.

Oxidative Stress Tools	Inflammatory Tools	Nutraceutical Compounds Tools	Motor and Cognitive Scales
SOD	Lipid profile	AX/AX + Sesamin	IPAQ
TBARS	IL1/IL1b/IL6 /IL-8/IL-10 IL-12	Beetroot juice (Nitrate dietary) NO^3−^	MET
FRAP	I CAM-3	Vitamin C	HOMA-IR
ROS/iNOS ROM	E/P-selectin Adhesion molecules		VAS/NPRS
TOS/TOC TAS/TAC/TACV	Thrombomodulin Endothelin-1	Capsinoids	1RM
3-NT	TUBB	Isoflavone	POMS2/TMD
CP	CRP	Cryotherapy	GDS/S-GDS
MDA	TNFα	N3 PUFA	BDHQ
GPx SOD/GPx ratio	HCY	European/Spanish/ Majorcan/ dietary intake	PASE
PCOOH	BNDF	Vitamins A/C/D Tocopherol	PHQ-8
CAT	8-iso PGF2α	Beta-carotene	PSQI/OSA-MA
GSR	IL-12/IL-10 ratio /IL-12-p70/ IL-8/IL-10 ratio	Acrylamide	MMSE/ MoCA
HSPs	Ferritin	Cocoa beverage powder	DS/DSF /DSB
H_2_O_2_	Fibrinogen	L-glutamine	TMT/ATMT/CDT
BAP		Ozone	SF-36-HRQOL
TRX		Vitamin E/Zn/Se	RAPA/GPAQ
GRd		Protein intake	EQ-5D
AOPP		Niacin	COWAT
GSH/GSSG		FST	UPDRS/H&Y /PDQ
NO		PM-EE	LOTCA
8-OHdG		Melatonin	D-KEFS–CWI
		H_2_/Photo-modulation	RAVLT/ CVLT/AVLT /LMT/SDMT
		Benfotiamine	IADL/ADL
		Vitamin B	WHODAS/DHQ/FFQ Er-Med
		NAC	FFQ
		Ladostigil	BDI/BAI
		Polyphenols/ Mind foods	BBS
		Kefir Probiotics	EDSS /SEP-59/MSQOL/
		LF	ADAS–cog/ ADCS–ADL/DAD/ CERAD-K/CDR
		CC	NPI/RBANS/NTB

Oxidative stress markers: SOD—superoxide dismutase enzyme; TBARS—thiobarbituric acid-reactive substances; TOC—total oxidative capacity; TAC—total antioxidant capacity; TACV—total antioxidative capacity; FRAP—plasma ferric reduction capacity; 3-NT—3 nitro-tyrosine; iNOS—nitric oxide synthase; CP—protein carbonyls; MDA—malondialdehyde; GPx—glutathione peroxidase; SOD/GPx ratio—stress score; PCOOH—plasma phosphatidylcholine hydroperoxide; CAT—catalase; GSR—glutathione reductase; HSPs—heat-shock proteins; H_2_O_2_—hydrogen peroxide; BAP—biological antioxidant potential; TRX—thioredoxin reductase; GRd—glutathione reductase; AOPP—advanced oxidative protein products; GSH/GSSG—reduced/oxidised glutathione; NO—nitric oxide; 8-OH-dG—8-hydroxy-deoxyguanosine. Inflammatory markers: lipid profile—total cholesterol; HDL-cholesterol—high-density lipoprotein cholesterol; LDL-cholesterol—low-density lipoprotein cholesterol; triglycerides; IL-1/IL-1b/IL-6/IL-8/IL-10/IL-12—interleukin 1, 1b, 6, 8, 10, 12; ICAM-3—intercellular adhesion molecule; TUBB—tubulin beta class I; CRP—C-reactive protein; TNF—tumour necrosis factor α; HCY—homocysteine; BNDF—brain-derived neurotrophic factor; 8-isoPGF2α—8-isoprostaglandin F2α. Nutraceutical compounds: AX/AX + Sesamin—astaxanthin/astaxanthin + sesamin; n3-PUFA—omega-3-polyunsaturated fatty acids; FST—Laminaria japonica; PM-EE—PhytoMeal ethanol extract; NAC—N-acetyl cysteine; LF—lactoferrin; CC—Cosmos caudatus. Motor and cognitive scales: IPAQ—International Physical Activity Questionnaire; MET—physical activity score; HOMA-IR—homeostatic model for the assessment of insulin resistance; VAS/NPRS—visual analogue scale/numeric pain rating scale; 1RM—test–retest reliability of one repetition, maximum assessment of the strength capacity of individuals; POMS2/TMD—profile of mood status/total mood disturbance; GDS/S-GDS—geriatric depressive scale/short geriatric depressive scale; BDHQ—brief-type dietary history questionnaire; PASE—Physical Activity Scale for the Elderly; PHQ-8/PSQ—Patient Health Questionnaire 8, which evaluates depression status; PSQI—Pittsburgh Sleep Quality Index/Ogri–Shirakawa–Azumi Sleep Inventory; MMSE/MoCA—Mini Mental State Examination test/Montreal Cognitive Assessment; DS/DSF/DSB—digit span/digit span forwards/digit span backwards; TMT/ATMT/CDT—Trail-Making Test/Advanced Trail-Making Test/Clock-Drawing Test using visuospatial memory; SF36-HRQ—short-form health-related quality of life questionnaire with 36 items; RAPA/GPAQ—Rapid Assessment of Physical Activity Questionnaire/Global Physical Activity Questionnaire; EQ-5D—Euro quality of life, which includes five dimensions: self-care, usual activities, mobility, depression/anxiety, pain/discomfort; COWAT—Controlled Oral Word Association Test; UPDRS/H&Y/PDQ—Unified Parkinson’s Disease Rating Scale/Hoehn and Yahr/Parkinson’s Disease Questionnaire (39 items); LOTCA—Loewenstein Occupational Therapy Cognitive Assessment; D-KEFS–CWI—Delis Kaplan executive function system–colour and word interference; RAVLT/CVLT/AVLT (IR, SR, LR)—Rey Auditory–Verbal Learning Test/California Verbal Learning Test/Auditory Verbal Learning Test (immediate recall, short recall, long recall); LMT/SMDT—Logical Memory Test/Symbol Digit Modalities Test; IADL/ADL—instrumental activities of daily learning/activities of daily living; WHODAS/DHQ/FFQ/Er-Med—WHO disability assessment schedule-related dietary intake (protein, fibre, fruits, vitamins)/dietary history questionnaire/food frequency questionnaires/17-item questionnaire for Mediterranean foods; BDI/BAI—Beck Depression Inventory (score ranges 0–63, with 21 items)/Beck Anxiety Inventory; BBS—Berg Balance Scale; EDSS/SEP-59/MSQOL—expanded disability status score/59-item multiple sclerosis quality of life questionnaire; ADAS–cog/ADCS–ADL/DAD/CERAD-K/CDR—Alzheimer’s disease assessment scale–cognitive subscale/Alzheimer’s disease cooperative study–activities of daily living/disability assessment in dementia/consortium to establish a registry for Alzheimer’s disease/clinical dementia rating; NPI/RBANS/NTB—neuropsychiatric inventory/repeatable battery for the assessment of neuropsychological status/neuropsychological test battery.

**Table 2 antioxidants-12-01008-t002:** Beneficial effects through physical activity and nutraceutical dietary intake on measurement outcomes.

Authors	Design	Dietary Intake	Age/ Gender	Physical Activity	Primary Outcomes	Secondary Outcomes	Tools/ Instruments	Conclusions
Liu S.Z. et al. (2021) [23]	42 subjects: 23 females 17 males	Nutraceutical dietary (AX) AX group PL (placebo) group	65–82 years	12 weeks endurance training for 3 sessions/week	FATox (g/min) CHO (g/min) RER Exercise efficiency (%) = work (kcal/min)/energy expenditure (kcal/min) × 10	GXT, TA, blood chemistry test: lipid profile (total cholesterol, HDL, LDL, triglycerides insulin level)	Treadmill, custom-built exercise device	Improved exercise efficiency for males, enhanced FA tox for both sexes and groups, increased fat oxidation under the same exercise intensity, improved exercise efficiency, decreased carbohydrate oxidation during lower-intensity training in older males.
Oggioni C. et al. (2018) [26]	20 participants 10 females 10 males	Nitrate dietary (NO_3_^−^) beetroot juice: 70 mL × 2/day for seven days	60–70 years	Cycling with 10-M, pedalling cadence of 50 rpm, vigorous/moderate-intensity activity, walking, sitting	IPAQ, MET, haemodynamic parameters: BP, CO, CI, SV, HR, AIx	Glucose, insulin, HOMA-IR, IL-6, cGMP, E-selectin, P-selectin, thrombomodulin ICAM-3, 3-NT	Bicycle ergometer, micromanometer	The nitrate nutraceutical diet did not favourably influence the haemodynamic and cardiac parameters at rest or during physical activity of various degrees, nor the oxidative stress.
Żychowska M. et al. (2021) [17]	24 women SUP (supported group), CON (control) group	1000 mg vitamin C/day for 6 weeks	≥65 years	6-week training program three times/week 60 min, either session, gyrokinesis–Nordic walking, stabilisation training	VO_2_ max, muscle mass, fat mass, BMI, TOS/TOC, TAS/TAC vitamin C prooxidative/antioxidative ratio	TUBB, IL-1, IL-6, IL-10, CCL2, CRP	Cyclo-ergometer Ergoline (60 rpm cadence)	Decreased BMI and increasing muscle mass after regular exercise training-related CON group, insignificant changes for both groups after 6 weeks of exercise training supported/not supported with vitamin C, non-influencing oxidative/antioxidative balance after support using vitamin C and health training, decreased IL-6 and increasing IL-10 mRNA in supported group, small increase in IL-1 after aerobic training and nutraceutical vitamin C supplementation.
Yokoyama K. et al. (2020) [32]	69 subjects 52 females 17 males CP group PL group	9 mg capsinoids (*Capsicum anuum* L., CH-19 Sweet) daily for 12 weeks	52–87 years	VPA (≥6 METs), LMPA (1.5–5.9 METs), physical strength test, 10 m walking time	FAS METs, energy expenditure (Kcal/day = METs × time × body weight (kg) × 1.05	Mood profile: POMS2, TMD, LBM, BMI, % body fat, visceral fat index, muscle mass %, FFM (kg), VAS	Accelerometer with 3 axes, stadiometer, Scala Yamato with weight, activity meter	In terms of body composition, ingestion of capsinoids decreased waist circumference, body fat percentage and visceral fat index, increasing energy expenditure in sedentary participants. Homeostatic capacity regulation with decreased chilly sensation in adults 80 years or older. Enhanced energy expenditure, oxidative phosphorylation in muscles, insulin resistance and lipids’ use under conditions of increased amount and time devoted to LMPA.
Jéssica S. Giolo et al. (2018) [13]	32 females ISO group = 17 PL group = 15	100 mg/day isoflavone nutraceutical for 10 weeks intervention	50–70 years	Aerobic training, resistance training (7 exercises, 30 sessions, 10 weeks 50 min/session)	BMI, VE/O_2_ ratio, VE/CO_2_ ratio, 1RM	Total cholesterol, triglycerides, HDL, LDL, uric acid, HBA1c, IL-8, IL-1β, TNF, IL-10, SOD, TBARS, FRAP	Stadiometer Sanny, treadmill (5.5 km/hour)	Lipid profile and markers of oxidative stress were not influenced by isoflavone nutraceuticals and the association with aerobic or resistance exercise, enhanced IL-8 levels and reduced total cholesterol from aerobic and resistance training promotion.
Wiecek M. et al. (2021) [14]	RUN = 10 UTR = 10	WBC—24 exposures to −130° for 3 min in cryochamber (whole-body cryotherapy), 3 times/week, 7 days	53–56 years	3–5 times/week, 55–150 km at 2 marathons/year 10/6 METs—very hard/hard PA for RUN, 4 METs—moderate-intensity exercise for UTR	BMI, Height, Fat mass, % body fat, Haemoglobin (g/dL), Haematocrit (%), ESR (mm/h), Leucocytes (10^3^/µL), Platelets (10^3^/µL), Fasting glucose, HBA1c, Total cholesterol, Triglycerides, HDL, LDL, Total protein, Fibrinogen (g/L), AIP	iNOS, ADMA, 3-NTR, CRP, IL-6, IL-1β, IL-10, HCY	Bamet KN-1 cryogenic chamber, Stopwatch	Increased i-NOS levels in older adults after 24 WBC treatments, no change in the level of 3-NTR, which is an indicator of nitro-oxidative stress, stimulated lipid oxidation using IL-6 metabolic effect, decreased IL-6 levels in RUN group compared with UTR group.
Kunz H.E. et al. (2022) [35]	63 elderly n3-PUFA = 30, PL = 33 Male = 29 Female = 34	n3-PUFA diet for 6 months	71.5 ± 4.8 years	Free-living physical activity (MVPA), cardiorespiratory fitness (VO_2_ max), 1RM, leg endurance test	BMI, SMI, body fat (%) RBC-EPA RBC-DHA CRP, ESR, lean mass (kg), leg lean mass (kg)	ROS, jO2, jATP, ACR, Phe, Tyr, FSR	Waist-worn accelerometer, treadmill	Enhanced anabolic response after supplementation with n3-PUFA related to acute and resistance physical activity, physical mobility maintenance and strengthened skeletal muscles, increased resistance training and strength gain, improved ATP level (contraction energy) after n3-PUFA during physical training with decreased protein synthesis rate, involving anti-inflammatory status after n3-PUFA addition.
Aparicio-Ugarriza R. et al. (2018) [31]	429 older adults Females = 57% Males = 43%	Low fitness group (low PF), Medium fitness group (medium PF), High fitness group (high PF), PHYSMED project, dietary intake (European, Spanish or Majorcan food)	55–85 years	Chair stand test, aerobic endurance, dynamic balance test (8-foot up-and-go), 6-minute walk test, handgrip strength	BMI, TDW, FFM,	25(OH)D, total cholesterol, vitamin B12, folate, triglycerides, HDL-c/LDL-c, HCY, creatinine, urea, uric acid, glucose, total protein albumin, haemoglobin, haematocrit, RBC folate, Fe/FER	Handgrip dynamometer	Decreased vitamin B12 and triglycerides in blood concentration and increased HCY, TC, HDL, LDL and creatinine levels, especially in the low PF group.
Kawamura T. et al. (2021) [2]	873 participants Females—296 Males—577	Dietary intake with vitamin A, C, tocopherol, beta-carotene, WASEDA’S health study	50–65 years	PF: leg extension power, CRF (mL/kg/min)	BDHQ, BMI, LBM, body fat %, HR, blood pressure, VO_2_ max	Fasting glucose, HBA1c, TC, HDL-c, LDL-c, TG, TBARS, PC, F2-IsoP, vitamin A, vitamin C α-tocopherol, β-carotene	-	Decrease in oxidative stress by about 10% after physical fitness and nutraceutical intake (with vitamin A, C, tocopherol and beta-carotene) for elderly females and males.
Veronese N. et al. (2021) [18]	4436 participants Females—2578 Males—1858	Dietary acrylamide intake	61.3 years	20 m repeated 2 times, 400 m walking, chair stands repeated 5 times	FFQ, BMI	PASE	-	Poor physical performance was associated with higher acrylamide dietary intake.
Imai A. et al. (2018) [29]	22 volunteers 12—AS group 10—Placebo group Females—11 Males—11	AS food antioxidants (3 mg astaxanthin and 5 mg sesamin), visual display terminal task, ergometer task	About 60 years	Mental tasks—4 h/day, four sets of 30 min and four sets of 30 min using advanced TMT, physical tasks— 4 h/day with cycle ergometer, 4 weeks	VAS, CFQ, BMI, POMS2, OSA-MA	PCOOH	Bicycle ergometer, plethysmograph	Increased antifatigue effect after AS supplementation, PCOOH level augmentation during physical and mental tasks was reduced in AS.
Munguia L. et al. (2019) [33]	60 subjects, Flavonoid (F) group, nonflavonoid (NF) group but highly alkalinised, placebo group	Cocoa beverage powder once/day	55–70 years	12 weeks 6 min walking test, 30 min/day training	TUG, sit-up test, 2 min step-in-place test, glycaemia, TG, HDL-c, LDL-c, TG/HDL index	SMI, EQ-5D	Handgrip strength	Reduced oxidative stress through dietary intake intervention, improved functional and cognitive profile with physical activity and nutraceuticals from the flavonoid group.
Amirato G.R. et al. (2021) [27]	44 participants, exercising group (PE)—21, non-exercising group (NPE)—23	L-glutamine (Gln) nutraceutical (maltodextrin) 30 days	60–80 years	Endurance training, strength training, 60–75 min/session 3 times/week	FFT, TUG, 5XST, IPAQ, BMI	D-fructoso amine, insulin, GSH, GSSG, iron, uric acid, TBARs, average power of extensor and flexor knee muscles	Cycle ergometer	Gln administration stimulated gluconeogenesis and controlled glycaemia levels for older adults, increasing redox potential and metabolic immunological, cardiovascular, respiratory and neuromotor adaptations using endurance and strength training for elderly women.
Vyas C.M. et al. (2020) [37]	391 participants, non-Hispanic white—183, Black—110, Hispanic—98	Behavioural factors (cigarette smoking, alcohol consumption, depression) Nutraceutical dietary supplement with vitamin D and omega-3	Mean 67 years	Low, intermediate and vigorous physical activity (LMVPA) training (<5.5–29.9 > 30 MET hours/week)	METs PHQ-8, BMI	mtDNAcn	Bicycle	Lowered mtDNA copy number for smoking participants and negative impact, especially on Black population, not related to other lifestyle and behavioural factors for lower mtDNA copy.
Balmes J.R. et al. (2019) [21]	87 participants 65%–females 35%–males	Ozone-controlled chambers with different concentrations (0 ppb, 70 ppb, 120 ppb)	55–70 years	Moderate exercise training	BMI, CT, LDL-c, CRP, IL-6, 8-isoprostane, P-selectin, monocyte-platelets conjugated	Endothelin-1, 3-NT, GSTM1, fibrinogen	-	Decreased 3-NT levels and increased CRP, 8-isoprostane, ET-1 and IL-6 proinflammatory markers after ozone exposure to elderly. However, changes in serum levels of proinflammatory factors had no significant effect.
Busquets-Cortés C. et al. (2018) [22]	127 participants Female—66 Male—61 Inactive group—40 Intermediate group—41 Active group—46	European food, Spanish food, dietary intake with vitamin C, vitamin E, Zn, Se	55–80 years	Intermediate and active exercise training	BMI, body fat %	CAT, SOD, GRd, GPx, MDA, 3-NT, TrxR1, PBMCs	-	Increased antioxidative activity and attenuated inflammatory status-related aging due to an active lifestyle and regular physical activity promotion, improved oxidative metabolism capacity in PBMCs and enhanced antioxidant defences.
Ten Haaf D.S.M, et al. (2020) [30]	104 subjects 81%—male 19%—female protein group—50 placebo group—54	Protein dietary intake for 12 weeks with 2 nutraceutical proteins/day	67–73 years	30/40/50 km/day walking training (endurance exercises), 3 days, consecutive	BMI, NPRS	CK, CRP, IL-6, IL-10	-	Protein nutraceutical administered to healthy older adults did not affect muscle damage, soreness and fatigue after prolonged moderate-intensity walking training.
Batista R.A.B. et al. (2022) [38]	950 subjects 474—females 476—males	Glycohaemoglobin group (<5.7%), glycohaemoglobin group (>5.7%), dietary intake with omega-3 (2–4 weeks)	50–85 years	Moderate or vigorous physical activity, strength training	BMI	Fasting glucose, HBA1c %, plasma omega-3, ALA, EPA, DHA	Handgrip strength- isokinetic dynamometer	Increased plasma omega-3 level improved muscle strength in elderly people with normal glycaemia, decreased inflammation and oxidative stress related to physical activity training of different degrees.

AX: astaxanthin; PL: placebo; FA tox: fat oxidation; CHO: carbohydrate oxidation; RER: respiratory exchange ratio = VCO_2_/VO_2_; GXT: graded exercise test; TA: tibial anterior endurance test; IPAQ: international physical activity questionnaire; FFQ: food frequency questionnaire (energy intake kcal/day); BP: blood pressure; CO: cardiac output; CI: cardiac index; AIx: augmentation index related to % pulse pressure; SV: stroke volume; HR: heart rate; HOMA-IR: homeostatic model for the assessment of insulin resistance; IL-6,1,10: interleukin-6,1,10; cGMP: guanosine monophosphate; ICAM-3: intercellular adhesion molecule; 3-NT: nitro-tyrosine; TOS: total oxidative status; TOC: total oxidative capacity; TAC: total antioxidant capacity; TACV: total antioxidative capacity; BMI: body mass index; TUBB: tubulin beta class I; CRP: C-reactive protein; LMPA: light to moderate physical activity; FFM: fat-free mass; VAS: visual analogue scale; VPA: vigorous physical activity; FAS: fitness age score; METs: metabolic equivalents; POMS2: profile of mood status; TMD: total mood disturbance; LBM: lean body mass; VE/O_2_ ratio: ventilatory equivalents for oxygen; VE/CO_2_ ratio: ventilatory equivalents for dioxide; 1RM: test–retest reliability of one repetition maximum assessing the strength capacity of individuals; HDL: high-density lipoprotein; LDL: low-density lipoprotein; HBA1c: glycated haemoglobin; TNF: tumour necrosis factor; SOD: superoxide dismutase enzyme; TBARS: thiobarbituric acid-reactive substances; FRAP: plasma ferric reduction capacity; RUN: long-distance runners; UTR: untrained men; WBC: whole-body cryotherapy; iNOS: inducible nitric oxide synthase; ADMA: asymmetric dimethylarginine; 3-NTR: 3 nitro-tyrosine; HCY: homocysteine; ESR: erythrocyte sedimentation rate; AIP: atherogenic; n3-PUFA: omega-3 polyunsaturated fatty acids; MVPA: moderate to vigorous physical activity; SMI: skeletal muscle index; RBC: red blood cell; ROS: reactive oxygen species; jO2: oxygen flux; ATP: adenosine triphosphate flux; Phe: phenylalanine; Tyr: tyrosine; FSR: fractional synthesis rate; 25(OH)D: vitamin D; TG: total cholesterol; TDW: total body water; Fe: iron; FER: ferritin; RBC: red blood cell; PTPA: leisure-time physical activity; LOPCA: Loewenstein occupational therapy cognitive assessment; GDS: geriatric depression scale; BDHQ: brief-type dietary history questionnaire; CRF: cardiorespiratory fitness; PC: protein carbonyl; F2-IsoP: F2-isoprostane; PASE: Physical Activity Scale for the Elderly; TSDG: Taekwondo self-defence training course group; CG: control group; MDA: malondialdehyde; GPx: glutathione peroxidase; HST: handgrip strength test; CSST: chair sit-to-stand test; BST: back stretch test; V-SRT: sit and reach test; TUG: timed up-and-go test; 6-MWT: six min walk test; AS: astaxanthin and sesamin; CFQ: Chalder Fatigue Questionnaire; PCOOH: plasma phosphatidylcholine hydroperoxide; ATMT: Advanced Trail-Making Test; OSA-MA: Ogri–Shirakawa–Azumi Sleep Inventory; 5XST: five-time sit-to-stand test; RT: resistance training; CAT: catalase; GSR: glutathione reductase; TMPT: Trail-Making Peg Test; PHQ-8: Patient Health Questionnaire 8; mtDNA cn: mitochondrial DNA copy number; GSTM1: glutathione S-transferase Mu 1; TrxR1: thioredoxin reductase; UCP3: uncoupling protein 3; PBMCs: peripheral blood monocellular cells; LTPA: moderate leisure time physical activity; NPRS: numeric pain rating scale (0–10); EPA: eicosapentaenoic acid; DHA: docosahexaenoic acid; ALA: alpha linolenic acid.

**Table 3 antioxidants-12-01008-t003:** Physical activity and oxidative stress outcomes.

Authors	Participants	Characteristics	Age/Gender	Physical Exercises	Primary Outcomes	Secondary Outcomes	Tools	Conclusions
Daimiel L. et al. (2020) [49]	6874 participants 48.5% females, 51.5% males	PREDIMED-plus trial PF—physical fitness (PF quartiles) Chair stand test PA—physical activity (PA levels) Rapid assessment physical activity questionnaires	60–70 years	Chair stand test (30 s), light physical activities, moderate physical activities (≤150 min/week), vigorous physical activities (≤75 min/week), high PA (>150 min/week)	BMI, Er-Med diet, MMSE, COWAT, DS, TMT, CDT	SF36-HRQL, RAPA, MET/min/week	Chair stand, Stopwatch	Enhanced scores for verbal and phonemic verbal fluency related to PF, decreased TMT time in PF, improved neurocognitive parameters related to PF, increased cognitive function and quality of life in activities correlated with PF and PA.
Alghadir A.H. et al. (2021) [17]	106 individuals 44 females 62 males	Sedentary group (n = 29), moderately active group (n = 37), highly active group (n = 40)	56–81 years	Active exercises LTPA	VO_2_ max, RER, HCY, Vitamin E, NO, TAC	GPAQ, METs, LOTCA, MMSE	Treadmill, Accelerometer	Decreased glycosylated haemoglobin and homocysteine levels related to moderate and high physical activity, lowered vitamin E levels related to degrees of cognitive capacity damage, increased serum levels of oxidative stress markers and α- and γ-tocopherol at higher-intensity physical activity, and better cognition capacity.
Netz Y. et al. (2021) [50]	122 subjects	Younger elderly female (<74 years) elderly female (≥75 years) Males (n = 36) PA, PF	65–82 years	PA: habitual physical activity, low-, moderate- or high-level exercises, PF	BMI, IPAQ-SF, MMSE, METs, PSQI, GDS	FI, VO_2_ max, H_2_O_2_	Treadmill	The correlation of FI with oxidative stress markers, BMI, physical and fitness activities and sleep disorders suggested an improvement in motor function, mental and emotional status and sleep quality in elderly men and younger elderly women.
Ku B.J. et al. (2021) [19]	16 women TSDG—8 Control G—8	Postmenopausal status	>45 years	12-week Taekwondo training course, 4 times/week, session—60 min	BMI, fat mass (kg), percent fat (%), LBM	MDA, SOD, IL-6, α-TNF	Wear M430 device	Decreased MDA and increased SOD after Taekwondo training course, reduced oxidative stress after physical activity, decreased IL-6 and TNF blood levels, improved agility and motor functions for menopausal women after intervention of Taekwondo training course.
Morucci G. et al. (2022) [16]	18 participants 14—female 4—men	Functional fitness, reactive oxygen metabolites, biological antioxidant potential	62–86 years	24-week multimodal exercise program, 1 h/session, twice/week	BMI, body fat % FFTs (HST, CSST, BST, V-SRT, TUG, 6-MWT)	Salivary cortisol levels, ROM, EuroQol-5 dimension-3 level	Chairs, Elastic bands, Sticks, Electronic hand dynamometer	Increased flexibility for upper and lower limbs after multimodal exercise training application, fitness capacity maintenance from daily living activities, decreased body fat percentage improved muscle strength, aerobic endurance, dynamic balance and biomechanical and physiological parameters, cognitive function amelioration due to intense physical activity, which stabilised anti- and pro-oxidative balance status.
Takahashi M. et al. (2019) [52]	38 participants	Active group (PA)—19 Control group—19	70.2 ± 3.9 years	8 weeks MVPA, ADL	BMI, METs systolic blood pressure, diastolic blood pressure	GDS, step count (steps/day) BDNF, serotonin, ROMs, HEL, BAP, TRX	Uniaxial accelerometer	Increased step count, BDNF, serotonin levels after MVPA, depression prevention associated with decreased serotonin concentrations in older women due to PA, decreased oxidative stress, which promotes depression in older adults, using PA.
Mesquita P. et al. (2019) [11]	13 Participants (males)	Muscle biopsies, HSPs	64 ± 9 years	6 weeks RT twice/week, 3 sets with 10–12 repetitions	mRNA analysis, FFM	CAT, GDS, SOD, GAPDH	Isokinetic dynamometer	Benefit effect promotion in redox status, especially for CAT and GDS in skeletal muscles after RT training.
Yoon J. et al. (2020) [56]	24 participants, Synap group—15 Control group—9	Synapsology (SYNAP), Dual-task exercise using traditional games	65–77 years	8 weeks, 2 times/week, 60 min/session	BMI, TUG, 5XST, 5 m habitual test,	TMPT, ROMs, BNDF,	Game-like dual-task	Improved motor and cognitive abilities for older adults who participated in intervention games such as dual-tasks.
Valado A. et al. (2022) [24]	37 participants, experimental group—27, control group—10	Therapeutic pool with aquatic exercise training	60–89 years	hydrotherapy exercises, 15 sessions, 30 min/session	SOD	GPx, GR	-	Decreased bone damage risk due to aquatic exercise protocol application, increased enzyme-related antioxidant effects, hydrotherapy stimulated antioxidant defence for older adults.
Sánchez-Rodríguez M.A. et al. (2021) [1]	177 subjects (women)	Mexican community-dwelling women, behavioural factors	46–69 years	PA (walking, aerobics exercises, swimming, yoga, running) > 30 min/day	BMI, glucose, cholesterol, triglycerides, HDL-c	MDA, GPx, SOD, uric acid, SOD/GPx ratios, SS (stress score)	-	Increased oxidative stress index associated with sedentary lifestyle, smoking and coffee and alcoholic beverage consumption.
Kozakiewicz M. et al. (2019) [12]	327 participants Younger elderly inactive (YN)—112, younger elderly active (YA)—112, older inactive (ON)—128, older active (OA)—41	Younger elderly—65–74 years, Older—90–99 years	65–99 years	LTPA	BMI, METs, SDPAR, TC	MDA, SOD, CAT, GPx GR, SOD/GPx ratios, CP, isoprostanes	-	Increased GPx and CAT activity in both younger elderly and the oldest groups related to moderate physical activity, enhanced SOD/GPx ratios in younger elderly men compared with inactive older men of all groups, promoting moderate physical exercises caused decreased oxidative stress and had a beneficial effect on both younger elderly and the oldest groups.

Er-Med diet: 17-item questionnaire for Mediterranean foods; MMSE: Mini Mental State Examination test; COWAT: Controlled Oral Word Association Test; DS: digit span test; TMT: Trail-Making Test; CDT: Clock-Drawing Test using visuospatial memory; SF36-HRQL; Short-form health-related quality of life questionnaire with 36 items; RAPA: Rapid Assessment of Physical Activity Questionnaire; HBA1c: glycated haemoglobin; EPA: eicosapentaenoic acid; DHA: docosahexaenoic acid; ALA: alpha linolenic acid; GPAQ: Global Physical Activity Questionnaire; LTPA: leisure-time physical activity; FI: frailty index; PSQI: Pittsburgh Sleep Quality Index; GDS: Geriatric Depression Scale; H_2_O_2_: hydrogen peroxide; MDA: malondialdehyde; GPx: glutathione peroxidase; FFTs: senior functional fitness tests; TUG: timed up-and-go test; 6-MWT: six min walk test; 5XST: five-time sit-to-stand test; BDNF: brain-derived neurotrophic factor; HEL: hexanoyl lysine; BAP: biological antioxidant potential; TRX: thioredoxin; CAT: catalase; GSR: glutathione reductase; RT: resistance training; GAPDH: glyceraldehyde 3-phosphate dehydrogenase; GRd: glutathione reductase; TMTP: Trail-Making Peg Test; SDPAR: seven-day physical activity recall; CP: protein carbonyls.

**Table 4 antioxidants-12-01008-t004:** Relationship between dietary supplementation, physical activity and oxidative stress.

Authors	Participants	Characteristics	Age/Gender	Physical Exercises	Primary Outcomes	Secondary Outcomes	Tools	Conclusions
Gaitán J.M. et al. (2021) [36]	23 subjects: 50% female	Dietary nutraceuticals, UPA group, EPA group	60–69 years	26 weeks, aerobic exercise training, 150 min/week UPA, EPA, 3 sessions/week	VO_2_ peak CTSB, BDNF, Klotho protein, MMSE, CVLT	TC, TG, HDL-c, LDL-c, Non-HDL-c, lipid metabolites, nonlipid metabolism	Treadmill, accelerometer,	Increased CTSB associated with EPA, ameliorated cognitive function associated with aerobic training, decreased BDNF during physical activity, cardiorespiratory fitness associated with Klotho, enhanced phospholipids and PUFAs in the EPA group.
Malek Rivan N.F. et al. (2019) [40]	815 participants: cognitively pre-frail group (37.4%), cognitively frail group (2.2%),	LRGS-TUA study, questionnaire–interview low niacin intake	60 years	ADL, IADL, physical fitness,	BMI, MMSE, GDS, PASE, digit span, RAVLT, MoCA, MOS-SS, WHODAS, DHQ	SOD, MDA, DNA damage, vitamin D, BDNF, telomerase	Handgrip, chair	Decreased ADL associated with cognitive impairments and physical frailty, lowered niacin intake associated with enhanced risk of cognitive frailty in dementia, oxidative stress markers such as MDA and telomerase levels being suggestive of cognitive frailty in older adults.
Silva L.A.D. et al. (2019) [51]	Depression group—16 subjects, Non-depression group—14 subjects	Pool with depth of 1.20 m, 25 m × 12.5 m, water with temperature about 26 °C, dietary intake protein	63.5 ± 8.8 years	12 weeks of aquatic training, 2 times/week, 45 min/session	BMI, HR, Borg scale, BDI, BAI, TUG, BBS	GSH, SOD, nitric oxide, protein carbonylation	Stopwatch, ruler, chair	Aquatic training program contributed to decreasing anxiety, depression and other cognitive impairments, improved physical functions and SOD/GSH levels in the non-depression group, lowered oxidative stress after undergoing aquatic training.
Reid S.N.S. et al. (2019) [43]	60 participants: FST group—32, control group—28	Nutraceutical dietary intake with fermented Laminaria japonica (FST) 1.5 g/day for 6 weeks.	67–81 years	Physical fitness	MMSE, numerical memory test, Raven’s test, Flanker test, iconic memory test, TMT, 6-MW, TUG	SOD, GSR, GPx, TBARS, 8-oxoDG, BDNF	Stopwatch, armchair	Increased cognitive functions after supplementation with FST, enhanced antioxidant activity for enzymes such as: SOD, GSR, GPx, decreased oxidative stress markers’ (TBARS and 8-oxoDG) levels after FST administration for 6 weeks, increased BDNF levels associated with neuromuscular integrity and physical functions after FST nutraceuticals.
Zaenker P. et al. (2018) [55]	26 participants: female—19, male—7	EDSS 0–5 Dietary intake	about 54 years	12 weeks, high-intensity training—one session/week, resistance training—one session/week.	VO_2_ peak, HR, MTP	LET, QOL, SEP-59, MSQOL-54	Cycle ergometer, dynamometer, weights, elastic bands	Improved functional capacity for resistance and endurance training in multiple sclerosis, enhanced quality of life, especially for men with multiple sclerosis.
Lee W.J. et al. (2020) [42]	53 participants: PM-EE group—26, placebo group—27	Nutraceutical with PM-EE times 12 weeks, 600 mg/day	50–85 years	Regular exercises 2–3–4 times or every day	MMSE CERAD-K, K-CWST, S-GDS	ADAS–cog, memory, language, executive function	-	Improvement in K-CWST, ADAS–cog scales, enhanced cognitive task scores such as attention, perception, spoken language, working memory increased ADL abilities in AD patients.

CTSB: myokine cathepsin B; BDNF: brain-derived neurotrophic factor; UPA: usual physical activity; EPA: enhanced physical activity; VO_2_: cardiorespiratory fitness; CVLT: California Verbal Learning Test; D-KEFS–CWI: Delis Kaplan executive function system–colour and word interference; PUFA: polyunsaturated fatty acids; LRGS-TUA: LRGS towards useful aging (TUA); PASE: Physical Activity Scale for the Elderly; RAVLT: Rey Auditory–Verbal Learning Test; MoCA: Montreal Cognitive Assessment; ADL: activities of daily living, IADL: instrumental activities of daily living; MOS-SS: medical outcomes study social support survey; WHODAS: WHO disability assessment schedule-related dietary intake (protein, fibre, fruits, vitamins); DHQ: dietary history questionnaire; BDI: Beck Depression Inventory (score ranges 0–63, with 21 items); BAI: Beck Anxiety Inventory; BBS: Berg Balance Scale; EDSS: expanded disability status score; MTP: maxim tolerated power; LET: lactates at the end of training; SEP-59: 59-item multiple sclerosis and quality of life questionnaire; MSQOL-54: multiple sclerosis and quality of life 54-item questionnaire; SE: Salicornia europaea; PM-EE: PhytoMeal ethanol extract; CERAD-K: consortium to establish a registry for Alzheimer’s disease; K-CWST: Korean version of the colour–word Stroop test; S-GDS: short-form geriatric depression scale.

**Table 5 antioxidants-12-01008-t005:** Relationships between antioxidants and oxidative stress in neurodegenerative disease.

Authors	Participants	Characteristics	Age/Gender	Primary Outcomes	Secondary Outcomes	Tools	Conclusions
Jiménez-Delgado A. et al. (2021) [20]	26 participants with PD	Nutraceutical dietary melatonin—25 mg for 3 months, melatonin–placebo group, placebo–melatonin group	60–69 years	Lipoperoxides, nitric oxide metabolites, carbonyl groups, CAT	Mitochondrial complex I activity, mitochondrial respiratory control ratio		Decreased oxidative stress markers after supplementation with melatonin, increased complex I activity, catalase and respiratory control ratio.
Hong C.T. et al. (2021) [44]	18 patients with PD	Combination of molecular hydrogen (H_2_) and Photo-biomodulation therapy for 2 weeks	50–80 years	UPDRS part I, II and III H&Y (II/III)	Bun, creatinine, GOT, GPT, WBC, Hb, PLT	Light-emitting diode array	Decreased reactive oxidative species using H_2_ water intervention, stimulated mitochondrial functions, which increased ATP levels, alleviated cognitive function deterioration.
Coles L.D. et al. (2018) [41]	5 participants (3 female, 2 male) with PD	Nutraceutical with NAC 6000 mg/day for 28 days	54–73 years	UPDRS (I-III), H&Y,	NAC, Cys, GSH, GSH/GSSG, CAT, MDA, 4-HNE	MRS-Siemens Magnetom	Decreased oxidative stress markers after NAC administration, increased cysteine levels and implicit antioxidant capacity, such as GSH/GSSG and catalase.
Yoritaka A. et al. (2021) [54]	15 patients PD	Molecular hydrogen (H_2_) inhalation, 2 L/min, 2 times/day, 1 hour, for 16 weeks.	50–70 years	MDS-UPDRS, PDQ-39	N1, N8-dyacetylspermidin, 8-hydroxy-2-deoxiguanosine	MHG-2000α-H_2_ producing machine	H_2_ gas supplementation is sure and safe but did not beneficially influence PD patients.
Gibson G.E. et al. (2020) [46]	70 patients with AD, benfotiamine group—34, placebo group—36	Nutraceutical with oral benfotiamine for 12 months, 300 mg, twice/day	60 years	ADAS–cog, MMSE, METS, SRT, NPI, ADCS–ADL	CDR, FDG, thiamine (Th), AGE	PET (brain positron emission tomography)	Ameliorated cognitive and functional status in AD patients after administration of nutraceuticals with benfotiamine.
An Y. et al. (2019) [39]	MCI group—102, control group—68	Dietary nutraceutical intake with vitamin B, EMCOA study	50–70 years	MMSE, MoCA, SDMT, AVLT (IR, SR, LR) LMT, DSF, DSB, FFQ (33 items)	Vitamin B_6_, folate, vitamin B_12_, Hcy, ROS, MDA, 8-OHdG, 8-isoPGF2α	MALDI-TOF mass spectrometry	Decreased cognitive decline associated with vitamin B_12_ intake deficiency, adequate diet with folate and vitamin B_6_ was associated with better cognitive reserves, enhanced Hcy levels and oxidative stress markers in MCI patients with vitamin B deficiency.
You Y.X. et al. (2021) [53]	48 participants	Supplemented nutraceutical with CC 250 mg twice daily for 12 weeks	60–70 years	MMSE, digit span, RAVLT, VR, POMS	BNDF, COX-2, GSH, MDA, iNOS, SOD		Improved global cognitive function after CC supplementation for 12 weeks, ameliorated mood status after CC, decreased lipid peroxidation and implicit oxidative stress markers.
Schneider L.S. et al. (2019) [47]	210 participants, ladostigil group—103, placebo group—107	Addition ladostigil 10 mg/day for 3 years	55–85 years	CDR (score-0.5), MMSE (>24), WMS-RC (≤18),	NTB, DAD, RAVLT, GDS (>5), medial temporal lobe atrophy scale (>1)		Ameliorated neurodegeneration with ladostigil in mild cognitive impairments after administration at a low dose.
Clark D.O. et al. (2019) [34]	180 participants, MINDspeed study	MindFoods dietary intake (high-polyphenol food) and speed training (playing control games) for 12 weeks	60–69 years	FFQ, MMSE, Trail-Making Test, GDS, RBANS	Proinflammatory cytokines, anti-inflammatory cytokines, TBARS, mRNA	iPad, BrainHQ program	Decreased proinflammatory cytokines and oxidative stress markers after intervention with dietary intake of MindFoods and MINDspeed (cognitive training), enhanced anti-inflammatory cytokine levels after administration of nutraceuticals for patients with AD.
Ton A.M.M. et al. (2020) [28]	13 participants: women—11 men—2	Probiotic kefir nutraceutical dietary intake (2 mL/kg/day) for 90 days	78 ± 7 years	MMSE, immediate memory test, delayed memory test, Cookie Theft Picture Test, similarity test, Boston Naming Test, TMT-A, Clock-Drawing Test	IL-6, IL-1b, TNF-α, IL-8, IL12p70, IL-10, IL-8/IL-10 ratio, IL-12/IL-10 ratio, ROS, AOPP, NO, PARP-1	Recall board, picture	Improved cognitive functions, such as memory, visual–spatial function, language abstraction and conceptualisation skills, decreased oxidative stress markers associated with ROS, Increased NO levels and implicit mitochondrial activity enhancement due to lowered membrane potential and improved antioxidant capacity.
Mohamed W.A. et al. (2019) [15]	50 participants: female—22, male—28, LF group, placebo group	Supplemented with lactoferrin (LF) 250 mg/day for 3 months	60–85 years	MMSE, CDR, ADAS–cog 11	PI3K, p-Akt, Ach, 5-HT, MDA, NO, GSH, TAC, IL-6, IL-10, p-tau, Aβ42, caspase-3, HSPs, cholesterol		Decreased A beta 42, cholesterol, proinflammatory and oxidative stress markers after LF supplementation for 3 months, lowered heat-shock protein, caspase-3, and p-tau upon intake, ameliorated cognitive functions for patients with AD due to decreased inflammation and oxidative status.

PD: Parkinson’s disease; RCR: mitochondrial respiratory control ratio; UPDRS: Unified Parkinson’s Disease Rating Scale; H&Y; Hoehn and Yahr; ATP: adenosine triphosphate acid; NAC: N-acetylcysteine; Cys: cysteine; 4-HNE: 4 hydroxynonenal; MRS: magnetic resonance spectroscopy; MDS: Movement Disorder Society; PDQ: Parkinson’s disease questionnaire (39 items); DiAcSpd: N1, N8-dyacetylspermidine; ADAS–cog: Alzheimer’s disease assessment scale–cognitive subscale; CDR: clinical dementia rating; FDG: fluorodeoxyglucose; AGE: advanced glycation end products; METS: memory evaluation treatment service; SRT: the Buschke selective reminding test; NPI: neuropsychiatric inventory; ADCS–ADL: Alzheimer’s disease cooperative study–activities of daily living; MCI: mild cognitive time of light impairment; EMCOA: effects and mechanism investigation of cholesterol and oxycholesterol on AD; SDMT: symbol digit modalities test; AVLT (IR, SR, LR): Auditory–Verbal Learning Test (immediate recall, short recall, long recall); LMT: logical memory test; DSF: digit span forwards; DSB: digit span backwards; WMS-RC: Wechsler memory scale revised, Chinese version; 8-isoPGF2α: 8-iso prostaglandin F2α; MALDI-TOF: matrix-assisted laser desorption ionisation time of flight; CC: Cosmos caudatus; VR: visual reproduction; POMS: profile of mood state; COX-2: cyclooxygenase-2; iNOS: inducible nitric oxide synthase; NTB: neuropsychological test battery; DAD: disability assessment in dementia; RBANS: repeatable battery for the assessment of neuropsychological status; AOPP: advanced oxidative protein products; PARP-1: Poly ADP-ribose polymerase 1; PI3k: phosphatidylinositole-4,5 biphosphate 3-kinase; p-Akt: protein kinase; ACh: acetylcholine; 5-HT: serotonin; Aβ42: amyloid β; HSP: heat-shock protein.

## Data Availability

Not applicable.

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
