# Peer review of "New Directions to Approach Oxidative Stress Related to Physical Activity and Nutraceuticals in Normal Aging and Neurodegenerative Aging"

_antioxidants, 2023, doi:10.3390/antiox12051008_

Round 1
Reviewer 1 Report
This manuscript reviews the recent results on the detrimental effects of oxidative stress on aging, cognitive impairment, and neurodegeneration, and suggests exercise training and nutraceuticals as potential therapeutic interventions to reduce oxidative stress and improve antioxidant capacity. Although this study reviews many recent studies, it lacks new insights. There are many grammar errors throughout the manuscript which make it hard for readers to read.
1. This paper is not well-prepared. It requires substantial language editing.
2. Check the grammar of lines 12-14.
3. Check the grammar of lines 72-73.
4. The writing has a serious problem. Line 97.
5. You have used the abbreviation ROS in line 87: production of oxygen and nitrogen reactive species (ROS/RNS). But you used the abbreviation again in line 97: reactive oxygen species (ROS). These two (ROS) do not represent the same word.
6. In the first column (authors column) of Tables 3-5, what does the number before the authors mean? Eg, “5. Daimiel L., et al. (2020)” in Table 3, what does “5.” mean?
7. Lines 865-867. “In our study, were investigated two directions through interventions of physical activity on different patterns and supplemented with natural and artificial nutraceutical compounds which delay or limit these effects.” “were” should be “we”. You said in this study, you investigated two directions. But you only cite the papers in the literature and did not investigate this.
Author Response
Thank you for help us to improve the paper.

Reviewer 2 Report
This is an interesting review which outlines antioxidant strategies to prevent premature aging and progression to neurodegenerative disease. I just have a few comments for the authors,
1. I think the title needs to be changed as it is a bit vague and needs to be more specific.
2. The abstract lacks specifics and needs to be rewritten to improve clarity.
3. There is a lot of information in this review and I think the manuscript would benefit from the removal of the some of the sections as it could could with outlining the causes of oxidative stress, treatments and methods of assessing oxidative stress.
4. I would have liked clearer information on the mechanisms of action of the nutraceuticals outlined in this review and a rationale for their choice.
5. A clear therapeutic strategy disseminated from the literature used to write this review on how to target aging/neurodegenerative disease would be good.
6. Figure 1 needs redrawing with a detailed legend as it`s quite unclear.
7. The authors indicate that exercise is an antioxidant when it is in fact associated with oxidative stress although it can be anti-inflammatory.
8. What about the peroxisome and aging, given it`s high antioxidant status?
9. What about the limited transport of antioxidant compounds through the blood brain barrier which will have a bearing on treatments for neurodegeneration?
10. What about treatments that enhance the mitochondrial respiratory chain as this is a major source of oxidative stress in aging and should be included in the review?
Author Response
Thank you for suggestions and support.

Round 2
Reviewer 1 Report
The authors have addressed some of my comments.
Reviewer 2 Report
The authors have addressed all my concerns and have amended their paper appropriately.